# Estimating Permeability of Porous Media from 2D Digital Images

Gang Lei [1,2], Tianle Liu [1,*], Qinzhuo Liao [3,*] and Xupeng He [4]

1   Faculty of Engineering, China University of Geosciences, Wuhan 430074, China; leigang@cug.edu.cn
2   Shenzhen Research Institute, China University of Geosciences, Shenzhen 518063, China
3   State Key Laboratory of Petroleum Resources and Prospecting, China University of Petroleum-Beijing, Beijing 112249, China
4   EXPEC, Advanced Research Center, Saudi Aramco, Dhahran 34258, Saudi Arabia; xupeng.he@kaust.edu.sa
*   Correspondence: liutianle2008@163.com (T.L.); liaoqz@cup.edu.cn (Q.L.)

**Abstract:** Digital rock physics (DRP) has been widely used as an effective approach for estimating the permeability of porous media. However, conventional implementation of DRP requires the reconstruction of three-dimensional (3D) pore networks, which suffer from intensive memory and underlying uncertainties. Therefore, it is highly significant to develop an approach only based on two-dimensional (2D) cross-sections of parent samples without 3D reconstruction. In this study, we present a novel approach that combines the Kozeny–Carman equation with fractal theory to derive a bridge function that links 2D cross-sectional images and 3D pore structures of parent samples in flow equivalence. Using this bridge function, we predicted the physical properties of the parent samples, including the permeability, bulk porosity, tortuosity, and pore fractal dimension. To validate our model, we performed Lattice Boltzmann (LB) simulations on nine carbonate samples and compared the LB simulation results with our model's predictions. We also compared our predicted results with available data on various porous materials, such as sandstone, glass beads, and carbonate, in the literature. Our findings demonstrate that without reconstructing 3D pore networks, our method provides a reliable estimation of sample permeability using 2D cross-sectional images. This approach not only simplifies the determination of sample permeability in heterogeneous porous media but also sheds new light on the inherent correlations between 2D cross-sectional information and 3D pore structures of parent samples. Moreover, the derived model may be conducible to a better understanding of flow in reservoirs during the extraction of unconventional onshore and offshore oil/gas.

**Keywords:** porous media; 2D digital image; 3D pore structures; permeability; bridge function

## 1. Introduction

Permeability, as one of the most fundamental physical properties of porous material [1–5], highly depends on the pore structure [6–9]. Estimating the permeability of porous material from its pore structure has attracted much attention in various environmental and energy applications, such as subsurface water management, geological $CO_2$ sequestration, geothermal energy extraction, and unconventional onshore and offshore oil and gas recovery processes [10–14].

Generally, there are three approaches for estimating the permeability of porous material, including experimental measurement, empirical models, and digital rock physics (DRP). Experimental measurement is time-consuming and expensive. Moreover, this approach is not repeatable due to its destructive feature. Instead, empirical models, including the well-known Kozeny–Carman (KC) equation and its variations, have been proposed by various researchers [7,10–12,15,16], of which permeability is related by several static parameters, such as porosity, formation factor, specific surface, and other pore structure parameters. Although these empirical models are efficient and cheap [17], they suffer from

general applicability due to the requirement of fitting parameters, which are generally rock-type-dependent [12,18].

As an alternative, digital rock physics (DRP) has been widely used [19–28]. However, conventional implementations of DRP require the reconstruction of three-dimensional (3D) pore networks, which suffer from intensive memory and underlying uncertainties. The underlying uncertainties could include resolution loss, structure distortion, non-uniqueness, etc. [26]. Moreover, high-fidelity numerical simulations performed on 3D pore networks are computationally expensive, which limits the general applicability of DRP.

To alleviate such limitations, many efforts have been made to directly estimate permeability from available 2D digital images by investigating the similarities between 2D pore structures and corresponding 3D ones [17,25,26,29,30]. Berryman and Blair [31] predicted the permeability of various porous materials by combining the pore information (porosity and two-point spatial correction function) from 2D SEM images with a form of the KC equation and validated their model with experimental data. However, experimental measurements of the formation factor in their model are missing. Yu and Chen [29] derived a fractal model by incorporating pore structure parameters (e.g., fractal dimension and the pore radius) from 2D images. However, direct extraction pore information from 2D images is different from 3D pore information of parent samples, which is supported by Sisavath et al. [32] and Saxena et al. [26]. Thus, transforming 2D pore information into 3D is necessary. Saxena et al. [26] compared rock permeability in 3D and the permeability computed directly from 2D slices and then proposed a bridge function between the 2D results and the 3D permeability. However, there are two fitting parameters (i.e., the normalized magnitude of variation in the pore radius and the "wavelength" of variation in the pore radius) in their model, which are difficult to be obtained. Recently, by combining the KC equation and Young–Laplace equation, Chen et al. [17] derived a bridge function to link the 2D pore structure and 3D pore structure and predicted the capillary pressure curve (CPC) of porous materials. It was found that their approach is useful to predict the CPC of porous media with a 2D pore structure. However, as they suggested, it was of great practical significance to propose other bridge functions to study the corresponding physical properties (e.g., thermal conductivity coefficient, acoustic conductivity) of porous media. To date, how to effectively transform 2D pore information from 2D digital images into 3D pore information without 3D reconstruction and then predict the 3D permeability of porous material still remains an open question and requires further investigation.

In this paper, we developed a novel, robust, and effective approach to predict the 3D permeability of porous materials from 2D images without 3D reconstruction. Previous studies reveal that interspaces in porous media possess fractal characteristics [12,29,33–37]. Specifically speaking, many scholars [12,29,35,37] suggested that the pore fractal dimension and tortuosity fractal dimension were two key parameters characterizing fluid flow in porous media. Thus, in this paper, these two parameters were applied to characterize the permeability of porous media. As the box-counting algorithm [38] and theoretical models [29,35] can effectively predict the pore fractal dimension [12,39–41], in this paper, the pore fractal dimension was estimated by the box-counting algorithm and theoretical models. In 2015, Wei et al [42]. derived a tortuosity fractal dimension model for fractal porous media that is related to porosity and the pore fractal dimension. As this model is effective at determining the tortuosity fractal dimension [43,44], this model was also used in this paper to model the tortuosity fractal dimension.

Recently, as a promising technique for modeling the fluid flow in porous media at the pore scale, the lattice Boltzmann Method (LBM) has been widely applied by scholars to predict properties (e.g., permeability and relative permeability) of porous media [45–49]. For example, Khodja et al. [46] stated that compared to the traditional methods of computational fluid dynamics, the LBM was ideally suited for massively parallel computation, and it had the advantages of simplicity and flexibility in dealing with complicated geometry. Thus, in this paper, the D3Q19 LBM model was applied to validate our derived model.

In what follows, a new analytical model was derived to transform 2D pore information into a 3D pore structure and give an estimation of the 3D permeability for porous media. Compared with DRP, these models will skip huge workloads of reconstructing 3D pore networks. Subsequently, lattice Boltzmann (LB) simulations were conducted on nine carbonate samples to validate our derived model. In addition, the available experimental data was applied to further validate the developed model and is followed by discussions of this model. Finally, the conclusions are presented.

## 2. Methodology

In this section, based on the pore morphology of 2D cross-sectional images, an analytical permeability of 3D global matrix pore structure is presented. It should be noted here that, for simplicity, all the 2D cross-sections are perpendicular to the flow direction; thus, the micro-fractures in actual porous media are ignored. The specific steps are as follows (Figure 1): Firstly, pore structure of 2D cross-sectional images will be characterized using fractal theory, and the permeability of 2D pore structure $K_{2D}$ will be derived based on the tube bundle model and fractal geometry [29,39,40]. In the capillary bundle model, the capillary bundles are perpendicular to the images (i.e., capillary bundles run along $y$ direction). Then, by combining the KC equation and fractal geometry, 3D pore structure of porous media in flow equivalence will be estimated, and the permeability of natural samples (in 3D) $K_{3D}$ will be derived. Finally, a bridge function connecting $K_{2D}$ and $K_{3D}$ will be proposed.

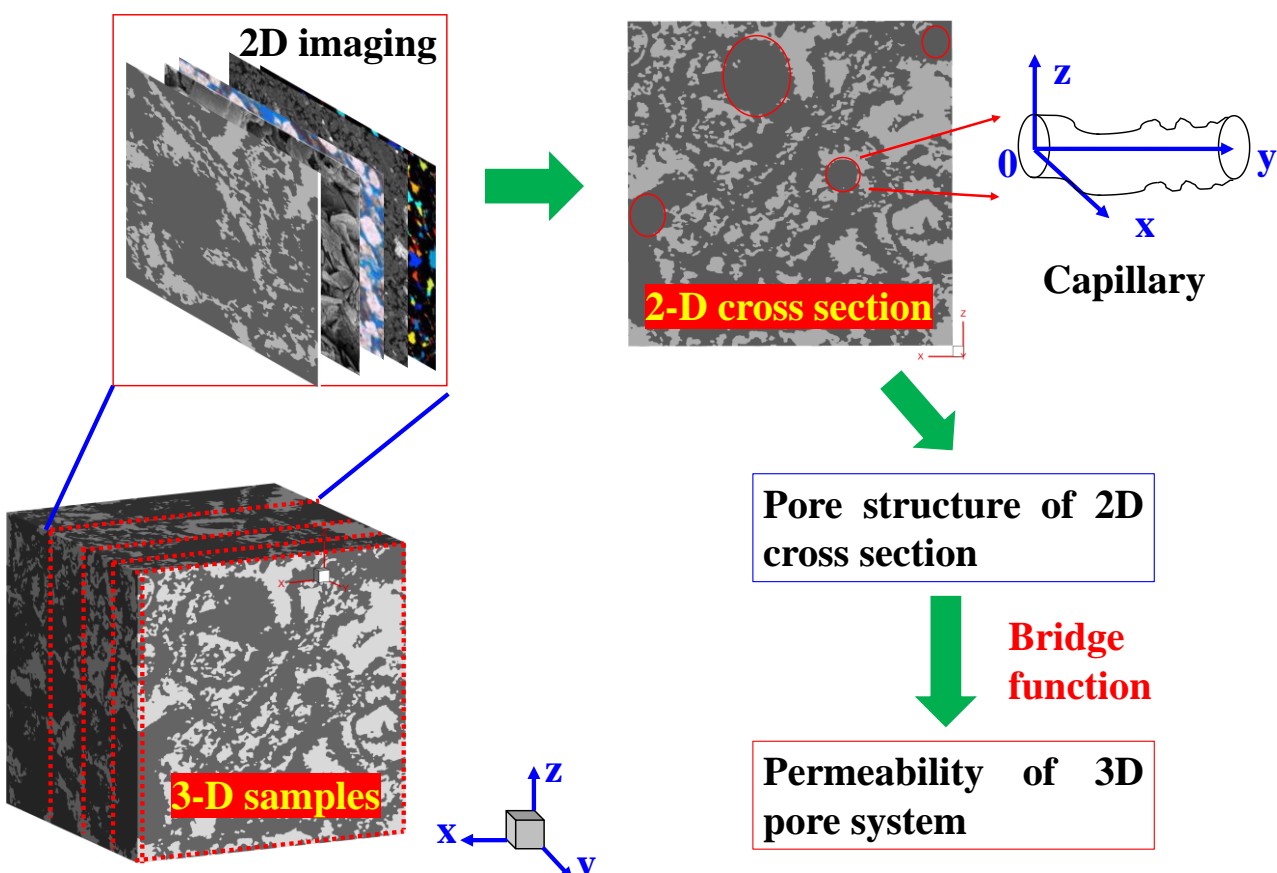

**Figure 1.** Schematic of transforming the 2D pore information into 3D pore structure and predicting 3D permeability from 2D cross-sections.

**Permeability of 2D cross-section:** In general, based on 2D imaging (thin section, back scatter electron, micro-CT scanning, scanning electron microscope, scanning electron microscopy, and electron probes, etc.) and digital image processing techniques, pore

structure parameters (areal porosity, pore size distribution, average pore radius, pore perimeter, specific surface area, pore fractal dimension, coordination number, and shape factor, etc.) of 2D cross-sections can be determined [26,31,50,51]. For instance, pore fractal dimension of 2D cross-section can be obtained by using the box-counting algorithm [38]. In addition, areal porosity of 2D cross-section can be obtained by thin section analysis or digital image processing techniques. Moreover, by using digital image processing techniques, various pore structure parameters (pore size distribution, the maximum pore size, minimum pore size, average pore radius, pore perimeter, and specific surface area) of 2D cross-sectional images can be easily determined [27,36,46,52,53]. For example, for a given 2D digital image, parameters ($r_{max}$, $r_{min}$, and $\varphi_{2D}$) can be determined by using the Avizo software, which can effectively determine pore space morphology and extract pore networks [25].

It is well-known that, affected by complex sedimentary and diagenetic processes, pore structure of natural porous media is complicated with obvious heterogeneity, which is difficult to be characterized. Specifically, many scholars suggested that Euclidean mathematics are not suitable to characterize a complex pore structure [29,33,34,54]. With the concept of "Fractal geometry" first proposed by Mandelbrot [33], the problem was well-solved. A large number of studies (experimental and theoretical studies) show that pore structure of most of the sedimentary rocks has obvious fractal phenomena. According to Equation (A4), by combining determined pore structure parameters (e.g., pore radius $r$, areal porosity $\varphi_{2D}$, pore fractal dimension $d_f$) from 2D imaging and fractal theory, permeability of 2D cross-section (i.e., 2D composite permeated with tortuous capillary tubes that are perpendicular to the 2D cross-section, Figure 1) $K_{2D}$ is given by [12,29,37,40,55]:

$$K_{2D} = \frac{\pi d_f r_{max}^{d_f} \left( r_{max}^{3-d_f+d_t} - r_{min}^{3-d_f+d_t} \right)}{2^{4-d_t} \sqrt{A^{1+d_t}} (3 - d_f + d_t)},\tag{1}$$

where the subscripts max and min denote the maximum and minimum, respectively. More details about Equation (1) can be found in Appendix A. In Equation (1), $d_t$ is tortuosity fractal dimension in 2D space, which is [42]:

$$d_t = (2 - d_f + 1) + (2 - d_f) \frac{\log(d_f) - \log(d_f - 1)}{\log(\varphi_{2D})}.\tag{2}$$

Generally speaking, tortuosity fractal dimension $d_t$ ranges from 1 to 2 in 2D space. When the matrix plane is filled with so highly tortuous capillary tubes, $d_t$ is equal to 2; however, for straight capillary tubes, $d_t$ is equal to 1, and on this condition, Equation (1) can be simplified as:

$$K_{2D} = \frac{\pi d_f r_{max}^{d_f} \left( r_{max}^{4-d_f} - r_{min}^{4-d_f} \right)}{8A(4 - d_f)},\tag{3}$$

Furthermore, $A$ in Equations (1) and (3) is the cross-sectional area of the representative elementary volume (REV) in 2D space, which can be calculated by Equation (A3) as:

$$A = \frac{\pi d_f r_{max}^{d_f} \left( r_{max}^{2-d_f} - r_{min}^{2-d_f} \right)}{\varphi_{2D}(2 - d_f)} = \frac{\pi d_f r_{max}^2}{\varphi_{2D}(2 - d_f)} \left( 1 - \frac{r_{min}^{2-d_f}}{r_{max}^{2-d_f}} \right).\tag{4}$$

Physically, $d_f$ in Equation (1) is the pore fractal dimension, which ranges from unity to 2 ($1 < d_f < 2$) in 2D pore space. Mathematically, $d_f$ can be determined by box-counting algorithm [38] or by the following equation [29,35]:

$$d_f = d_e - \frac{\ln(\varphi_{2D})}{\ln(r_{min}/r_{max})},\tag{5}$$

wherein the Euclidean dimension $d_e$ is 2 in 2D pore space. Equation (5) reveals that, under a given ratio $r_{min}/r_{max}$, $d_f$ increases with an increasing of areal porosity $\varphi_{2D}$. In addition, by fitting the experimental data, Chen et al. [56] also found a positive correlation between $d_f$ and $\varphi_{2D}$ and suggested $d_f$ could be estimated by $d_f = 0.974\varphi_{2D}^{0.173}$. Equations (1) through (5) form the basis for analysis of $K_{2D}$, which were utilized to estimate permeability of 3D parent porous media in flow equivalence.

**3D pore structure in flow equivalence:** As 2D cross-sections come from 3D parent samples, various studies show that pore structure information (pore morphology) of 2D slice is identical to that of 3D matrix system [17,30,57]. For example, Chen et al. [17] suggested that areal porosity of 2D cross-sections $\varphi_{2D}$ should be similar to bulk porosity of 3D matrix $\varphi_{3D}$. In addition, to estimate permeability in 3D from 2D images, Saxena et al. [26] also assumed that $\varphi_{2D}$ was similar to $\varphi_{3D}$. Figure 2a compares bulk porosity of 3D parent samples versus areal porosity of 2D cross-sections [17,26,31,36,46,53]. Taking the data from Saxena et al. [26], for example, Saxena et al. conducted modeling on 4 samples (i.e., a Fontainebleau sandstone, a Berea sandstone, a Bituminous sand sample, and a Grosmont carbonate). Based on digital rock technology, the bulk porosity $\varphi_{3D}$ for each sample was determined. Then, multiple 2D cross-sections (lying perpendicular to the flow direction) of a parent cube sample were sampled, and the corresponding areal porosity of each slice was measured using digital image process technology. Similarly, for 3D samples with bulk porosity determined, Wu et al. [36,53] divided the original sample into various layers along vertical direction using low- and high-resolution X-ray CT scanning technologies. Then, areal porosities in each layer of low- and high-resolution digital rock images were measured. Besides digital rock technology, thin section analysis was used by Berryman and Blair [31] and Chen et al. [17] to measure areal porosity of 2D thin. As shown in Figure 2a, porosity of 3D sample $\varphi_{3D}$ is approximately in the middle between the maximum and minimum values of areal porosities of 2D cross-sections. Specifically, for a 3D porous media with bulk porosity $\varphi_{3D}$, most of the areal porosity data $\varphi_{2D}$ for 2D sectional sections are in the range of $0.9\varphi_{3D}$ and $1.1\varphi_{3D}$. Figure 2b compares bulk porosity of 3D samples versus average porosity of 2D cross-sections. Results suggest that porosity of 3D sample is quite consistent with the average areal porosity of 2D slices. Specifically, the slope of the fitting line is 1.009, which is close to unity. And the intercept of the fitting line is close to $3.38 \times 10^{-4}$, which is quite close to 0. As a result, for simplicity of the model, areal porosity of 2D cross-sections $\varphi_{2D}$ is assumed to be identical to bulk porosity of 3D matrix (parent samples) $\varphi_{3D}$, i.e.,

$$\varphi_{3D} = \varphi_{2D}. \tag{6}$$

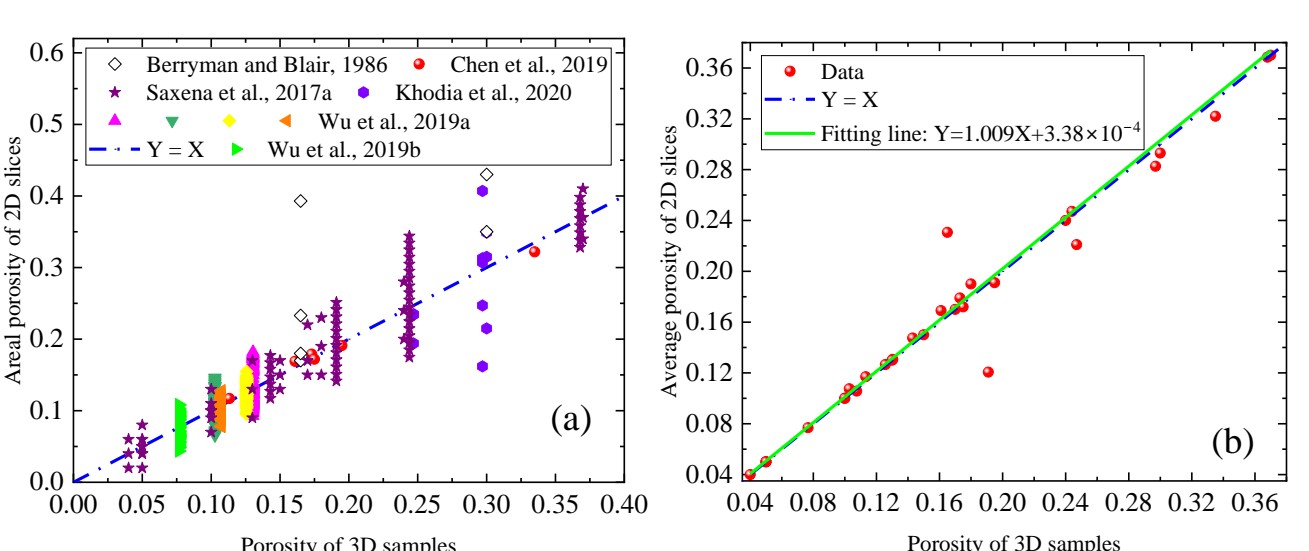

**Figure 2.** (**a**) Bulk porosity of 3D parent samples versus areal porosity of 2D cross-sections [17,26,31,36,46,53]; (**b**) porosity of 3D samples versus average porosity of 2D slices.

In general, for actual pore structure in 3D space, capillary tube is not straight and tortuosity $\tau$ (or tortuosity fractal dimension $D_T$) is larger than unity. Based on KC equation, Chen et al. [17] developed a bridge function connecting capillary radius in 2D space to that in 3D space. As stated by Chen et al. [17], for a given capillary with pore radius $r$ in 2D space, the corresponding pore radius $r_{3D}$ in 3D space is:

$$r_{3D} = \frac{r}{\tau^4(r_{3D})}. \tag{7}$$

As tortuosity $\tau$ is larger than unity, Equation (7) reveals that pore radius $r_{3D}$ in 3D space is less than that in 2D space, and the ratio of $r$ to $r_{3D}$ is tortuosity to the fourth power, i.e., $\tau^4$. With the same method as Chen et al. [17], but assuming that the ratio of specific surface in 3D space to that in 2D space is 3/2 and not the tortuosity, Saxena et al. [26] suggested that the ratio of $r$ to $r_{3D}$ is $9\tau^2/4$. To some extent, $9\tau^2/4$ can be considered a special case of $\tau^4$ (e.g., when $\tau$ is assigned as 3/2, $9\tau^2/4$ equals $\tau^4$), so, in this paper, to make our model more general, Equation (7) was used to correlate $r$ and $r_{3D}$.

According to fractal theory, tortuosity $\tau$ of pore radius $r_{3D}$ is [29]:

$$\tau(r_{3D}) = \left( \frac{\sqrt[3]{V}}{2r_{3D}} \right)^{D_T-1}, \tag{8}$$

where $V$ is the bulk volume of the REV. In light of Equation (A10), $V$ can be determined on basis of pore fractal dimension $D_f$ in 3D space as:

$$V = \left\{ \frac{\pi D_f r_{3D,max}^{3-D_T}}{4\varphi_{3D}(3-D_T-D_f)} \left[ 1 - \left( \frac{r_{3D,min}}{r_{3D,max}} \right)^{3-D_T-D_f} \right] \right\}^{\frac{3}{3-D_T}}. \tag{9}$$

By combining Equations (7) through (9), we have:

$$\begin{cases} r_{3D,max} = \dfrac{r_{max}}{\left[ \sqrt[3]{V}/(2r_{max}) \right]^{\frac{4D_T-4}{5-4D_T}}}; \qquad r_{3D,min} = \dfrac{r_{min}}{\left[ \sqrt[3]{V}/(2r_{min}) \right]^{\frac{4D_T-4}{5-4D_T}}} ; \\[3mm] \dfrac{r_{3D,max}}{r_{3D,min}} = \left( \dfrac{r_{max}}{r_{min}} \right)^{\frac{1}{5-4D_T}}, \end{cases} \tag{10}$$

where the subscripts 3D,max and 3D,min denote the maximum and minimum in 3D space, respectively. In what follows, subscript 3D presents in 3D space. Physically, $r_{3D,max}$ should be larger than $r_{3D,min}$, which means $D_T$ in Equation (10) should be less than 1.25. Based on fractal theory, in 3D pore space, $D_T$ is [42]:

$$D_T = (3 - D_f + 1) + (3 - D_f) \frac{\log(D_f) - \log(D_f - 1)}{\log(\varphi_{3D})}, \tag{11}$$

wherein $D_f$ is pore fractal dimension in 3D space, which can be also determined by [29,35]:

$$D_f = 3 - \frac{\ln(\varphi_{3D})}{\ln(r_{3D,min}/r_{3D,max})}. \tag{12}$$

More details about Equation (11) can be found in Equation (A8). By substituting Equation (10) into Equation (12), $D_f$ can be rewritten as:

$$D_f = 3 - \frac{(5 - 4D_T)\ln(\varphi_{3D})}{\ln(r_{min}/r_{max})}. \tag{13}$$

By combining Equations (11) and (13), we have:

$$D_{\text{T}} = \left[ \frac{(5-4D_{\text{T}})\ln(\varphi_{3\text{D}})}{\ln(r_{\min}/r_{\max})} + 1 \right]$$
$$+ \left\langle \frac{(5-4D_{\text{T}})\ln(\varphi_{3\text{D}})}{\ln(r_{\min}/r_{\max})} \cdot \left\{ \frac{\log\left[3 - \frac{(5-4D_{\text{T}})\ln(\varphi_{3\text{D}})}{\ln(r_{\min}/r_{\max})}\right]}{\log(\varphi_{3\text{D}})} \right. \right.$$
$$\left. \left. - \frac{\log\left[2 - \frac{(5-4D_{\text{T}})\ln(\varphi_{3\text{D}})}{\ln(r_{\min}/r_{\max})}\right]}{\log(\varphi_{3\text{D}})} \right\} \right\rangle. \tag{14}$$

By solving Equation (14), parameter $D_{\text{T}}$ can be determined. Then, based on Equation (13), $D_{\text{f}}$ can be determined. By substituting Equation (10) into Equation (9), we have:

$$V = \left\{ \frac{\pi D_{\text{f}} r_{\max}^{3-D_{\text{T}}}(2r_{\max})^{\frac{(4D_{\text{T}}-4)(3-D_{\text{T}})}{5-4D_{\text{T}}}}}{4\varphi_{3\text{D}}(3-D_{\text{T}}-D_{\text{f}})} \left[ 1 - \left(\frac{r_{\min}}{r_{\max}}\right)^{\frac{3-D_{\text{T}}-D_{\text{f}}}{5-4D_{\text{T}}}} \right] \right\}^{\frac{3(5-4D_{\text{T}})}{3-D_{\text{T}}}}. \tag{15}$$

By substituting Equation (15) into Equation (10), $r_{3\text{D,max}}$ and $r_{3\text{D,min}}$ can be determined. Then, according to Equation (A11), 3D permeability $K_{3\text{D}}$ of porous media in flow equivalence can be given by:

$$K_{3\text{D}} = \frac{\pi D_{\text{f}} 2^{D_{\text{T}}} r_{3\text{D,max}}^{D_{\text{f}}} \left( r_{3\text{D,max}}^{3-D_{\text{f}}+D_{\text{T}}} - r_{3\text{D,min}}^{3-D_{\text{f}}+D_{\text{T}}} \right)}{16 V^{\frac{1+D_{\text{T}}}{3}}(3-D_{\text{f}}+D_{\text{T}})}. \tag{16}$$

Figure 3 shows the flow chart for the 3D permeability determination. In light of our derived model, the methodology workflow is summarized as follows:

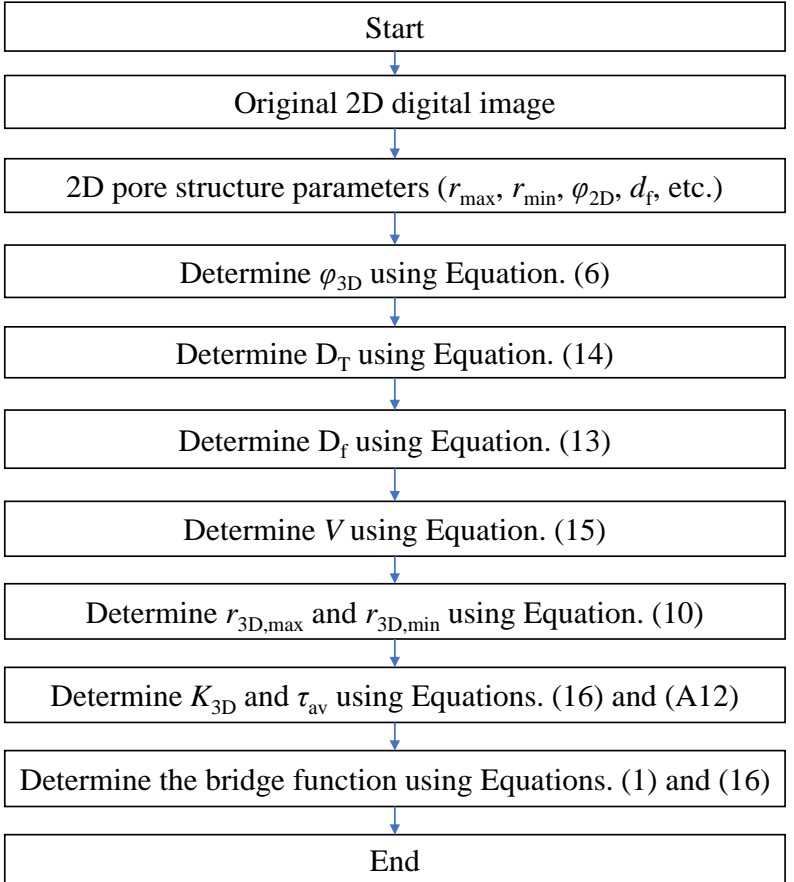

**Figure 3.** The flow chart for the 3D permeability determination from an original 2D digital image.

Step 1: Determine 2D pore structure parameters (e.g., $r_{max}$, $r_{min}$, $\varphi_{2D}$, and $d_f$) from 2D digital image. Specifically, for a given 2D digital image, parameters ($r_{max}$, $r_{min}$, and $\varphi_{2D}$) can be determined by using the Avizo software [25]. In addition, $d_f$ can be determined with box-counting algorithm [38] or with Equation (5). Subsequently, parameter $\varphi_{3D}$ can be calculated using Equation (6).

Step 2: Determine parameter $D_T$ by solving Equation (14). Then, determine $D_f$ by using Equation (13). In addition, determine parameter $V$ by using Equation (15). Then, $r_{3D,max}$ and $r_{3D,min}$ can be determined by using Equation (10).

Step 3: Determine $K_{3D}$ and $\tau_{av}$ of the corresponding parent sample using Equations (16) and (A12). In addition, the bridge function (i.e., the ratio of $K_{2D}$ to $K_{3D}$) can be determined by combining Equations (1) and (16). Moreover, other 3D pore structure parameters of the parent sample in flow equivalence can be further estimated.

## 3. Model Validation

Sampling and Experiments: 3 carbonate core plugs (labeled sample 1, sample 2, sample 3) with a diameter of 25 mm and thickness of 6 mm in the Middle East were selected. For these samples, multiple cubical sites were semi-randomly selected and then used in micro-CT X-ray imaging with the Versa XRM-500 X-ray micro-CT scanner and in simulation with the lattice Boltzmann method (D3Q19 model) to determine the permeability in 3D. Taking sample 1, for example, two cubical sites (site A and site B) were semi-randomly selected to avoid vugs. For site A with a bulk porosity $\varphi_{3D}$ of 23.5% and a total volume of $1.64^3$ mm$^3$, the volume (voxel) is $648^3$, and the voxel volume is $2.53^3$ μm$^3$. With the D3Q19 model stated in Appendix B, the determined permeability in 3D is $27.8 \times 10^{-3}$ μm$^2$. In addition, for site B with a bulk porosity $\varphi_{3D}$ of 23.4%, the volume (voxel) is $681^3$, the voxel volume is $1.55^3$ μm$^3$, and the total volume is about $1.06^3$ mm$^3$. Then, based on LBM simulation, the 3D permeability of site B was determined, with the value of $69.4 \times 10^{-3}$ μm$^2$. More detailed information about the D3Q19 model can be found in Appendix B. For the sake of simplification, site $J$ ($J$ = A, B, C, $\cdots$) from the sample $I$ ($I$ = 1, 2, 3) is labeled as sample $I$–$J$.

To verify the derived model in Section 2, a cross-section of each sample was sliced and used to estimate the permeability of the corresponding parent sample. Specifically speaking, firstly the 2D pore structure parameters (e.g., $r_{max}$, $r_{min}$, $\varphi_{2D}$, and $d_f$) from the 2D digital image were determined; then, based on our derived model, the 3D permeability of the carbonates was predicted. In addition, our predicted 3D permeability was compared to that from D3Q19 LBM to validate the feasibility and effectiveness of our derived model. The experimental data and modeling results are summarized in Table 1. As shown in Table 1 and Figure 4a, the areal porosity (2D porosity) of the cross-section consists of the bulk porosity (3D porosity) of the parent sample. The results (Figure 4a) suggest the slope and the intercept of the fitting line are 1.048 (close to unity) and $7.56 \times 10^{-3}$ (close to 0), respectively. Similar findings are shown in Figure 2. The results shown in Table 1 suggest that the predicted 3D permeability of each parent sample from the derived model exhibits excellent agreement with the experimental data (the results from the LBM simulations). Taking parent sample 1-A (Figure 4a), for example, in light of the image processing technology, the $\varphi_{2D}$ of the selected 2D cross-section (Figure 4b) is 31.24%. In addition, with the box-counting algorithm [38], the double logarithmic (Lg) plot of the box size (pixel) versus the total pore number is presented in Figure 4b; then, the pore fractal dimension $d_f$ (the negative of the slope) was determined to be 1.686. As shown in Figure 4b, the pore structure of the 2D cross-section of sample 1-A has a typical fractal scale. In our modeling, $d_f$ and $\varphi_{2D}$ were assigned as 1.686 and 31.24%, respectively. $r_{max}$ was assigned as 535.06 μm, and $\varphi_{3D}$ was assigned as 23.5%. Moreover, $r_{3D,max}$ and $D_f$ were assigned as 41.18 μm and 2.81, respectively. Then, based on the derived model, the predicted permeability $K_{3D}$ is $27.8 \times 10^{-3}$ μm$^2$, which is in agreement with that determined by the LBM simulation. Figure 4c presents the pore structure parameters (e.g., $\tau_{av}$, $r_{max}/r_{3D,max}$, and $D_T$) of the samples. As we can see from Figure 4c, the parameter $r_{max}$ is larger than $r_{3D,max}$, which is consistent with the results stated by Chen et al. [17]. For these 9 samples, the ratio

$r_{max}/r_{3D,max}$ ranges from 3.18 to 10.58. In addition, $D_T$ ranges from 1.10 to 1.15, and $\tau_{av}$ ranges from 2.19 to 4.69. The results (Figure 4c) also suggest that even from the same parent sample, the different sites behave strongly in heterogeneity. For instance, sample 1-A and sample 1-B come from the same parent sample 1; however, their physical properties vary greatly. Figure 4d presents the average tortuosity versus the ratio $r_{max}/r_{3D,max}$. The results (Figure 4c) show that there exists an approximately linear relationship between the ratio of the maximum pore radius in 2D space to that in 3D space ($r_{max}/r_{3D,max}$) and the average tortuosity of porous media. Physically, when the average tortuosity of porous media is unity, the corresponding ratio $r_{max}/r_{3D,max}$ approximately equals unity. As we can see from the fitting equation, when $\tau_{av}$ is assigned as unity, $r_{max}/r_{3D,max}$ is determined to be 0.82, which is close to unity, which verifies our model. The reason for the deviation may be caused by the computational error.

In general, carbonates generally contain clay minerals, and the interaction between clay minerals and fluids (e.g., water, gas, oil) will significantly affect the permeability of porous media [37,40]. For example, Lei et al. [37] derived analytical permeability to study the effect of clay swelling on the permeability of clay-rich argillaceous porous media. In addition, as the interaction between solid minerals (e.g., illite, montmorillonite, and kaolinite) and fluids in porous media (e.g., carbonate media) will change pore structures, the effect of the physical or chemical nature of solid minerals on the fluid flow in porous media is of great significance. However, the derived model in this paper does not consider the interaction between solid minerals and fluids. Thus, in our future work, the interaction between solid minerals and fluids will be taken into account to make our model more reasonable.

**Table 1.** Experimental data and modeling results.

| No. | Samples [a] | LBM | | | | Proposed Model | | |
|---|---|---|---|---|---|---|---|---|
| | | $\varphi_{3D}$ (%) | $K_{3D}$ ($10^{-3}$ μm²) | $\varphi_{2D}$ (%) | $d_f$ | $R_{3D,max}$ (μm) | $D_f$ | $K_{3D}$ ($10^{-3}$ μm²) |
| 1 | 1-A | 23.5 | 27.8 | 31.24 | 1.686 | 41.18 | 2.81 | 27.8 |
| 2 | 1-B | 23.4 | 69.4 | 21.35 | 1.450 | 43.15 | 2.79 | 69.4 |
| 3 | 2-A | 31.5 | 1450 | 35.71 | 1.576 | 64.04 | 2.78 | 1450 |
| 4 | 2-B | 21.5 | 216.7 | 26.44 | 1.683 | 70.31 | 2.79 | 216.7 |
| 5 | 2-C | 34.9 | 1743.1 | 38.55 | 1.611 | 60.24 | 2.79 | 1743.1 |
| 6 | 3-A | 30.7 | 906.9 | 34.97 | 1.735 | 89.07 | 2.82 | 906.9 |
| 7 | 3-B | 16.2 | 103.1 | 10.81 | 1.415 | 49.04 | 2.74 | 103.1 |
| 8 | 3-C [b] | 31.3 | 3049.4 | 36.97 | 1.631 | 61.83 | 2.76 | 3049.4 |
| 9 | 3-D [b] | 40.7 | 2660 | 36.74 | 1.636 | 60.20 | 2.82 | 2660 |

[a] The samples are the sites in the end trims cut off from the original samples in the literature [46]. [b] Samples 3-C and 3-D denote sites D and E of sample 3 in Table 1 of the literature [46].

**Dataset of Saxena et al. [26]:** To further verify our derived model, the predicted $K_{2D}$ and $K_{3D}$ from the new model were compared with available test data [26]. Saxena et al. conducted numerical experiments on 12 samples from various geologic formations [26]. For these 12 samples, samples 1 to 3 come from the same parent Berea sandstone and represent Berea sandstones A to C, respectively. In addition, samples 4 to 6 represent Bituminous sands A to C, respectively. Samples 7 to 9 represent Fontainebleau (Font) sandstones A to C, respectively. And samples 10 to 12 represent Grosmont carbonates A to C, respectively. During the numerical experiments, Saxena et al. first calculated the $K_{3D}$ of the parent samples based on 3D digital rocks using an LBM simulation; then, they sliced 2D thin sections along the flow direction from the parent samples and performed an LBM simulation to estimate the $K_{2D}$ of the thin section [26]. It should be noted that to calculate the $K_{2D}$ of the thin section, the pore structure was restructured by permeating the pores in

the thin section with straight tubes along the flow direction. That is, for the restructured porous media used to determine $K_{2D}$, the tortuosity fractal dimension $d_t$ was equal to 1. As a result, during our modeling, $d_t$ was assigned as 1, and Equation (3) was used to validate our derived model.

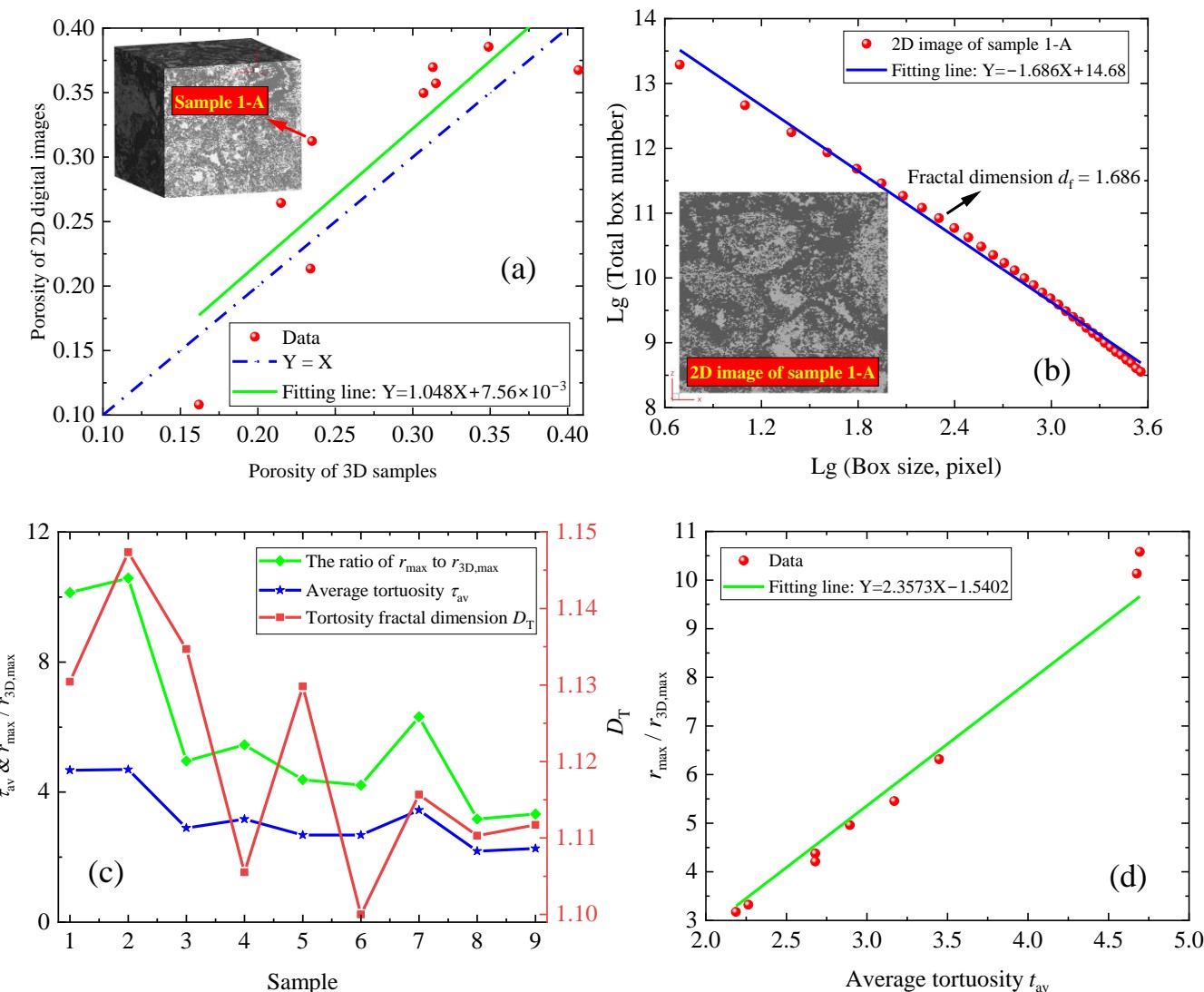

**Figure 4.** Modeling results of the samples: (**a**) the curves of $\varphi_{2D}$ versus $\varphi_{3D}$; (**b**) $d_f$ of the 2D image of sample 1-A; (**c**) the determined pore structure parameters for various samples; (**d**) $\tau_{av}$ versus the ratio $r_{max}/r_{3D,max}$ for various samples.

Taking sample 1 (sample Berea A) with a volume of $800 \times 800 \times 800$ µm$^3$ and a pixel size of 0.74 µm, for example, the bulk porosity $\varphi_{3D}$ is 4%, and the areal porosity $\varphi_{2D}$ of the thin sections ranges from 2% to 6% with an average of 4%. Based on simulations, the LBM 3D permeability of the parent sample was determined as $3 \times 10^{-3}$ µm$^2$, and the average LBM 2D permeability of the thin sections is $624 \times 10^{-3}$ µm$^2$. In our modeling, both $\varphi_{3D}$ and $\varphi_{2D}$ were assigned as 4%. Based on Equations (3) and (13), $K_{2D}$ and $K_{3D}$ can be calculated. The results (Figure 5a–d) suggest that the predictions ($K_{2D}$ and $K_{3D}$) provide a good match over the test data. Besides $K_{2D}$ and $K_{3D}$, other parameters (e.g., $d_f$, $\tau_{av}$, $r_{max}/r_{3D,max}$, $D_T$, and $D_f$) of each sample are presented in Figure 5e. As we can see from Figure 5e for these 12 samples, $\tau_{av}$ ranges from 1.10 to 1.50, which is smaller than that of the carbonates in Figure 4. This reveals that the carbonates in Figure 4 have stronger heterogeneity than the rocks in Figure 5. The reason may be that the rocks in Figure 4 are

carbonate core plugs; however, most of the rocks in Figure 5 are sandstones, which are more homogeneous. In addition, Figure 5e shows that the ratio $r_{\max}/r_{3D,\max}$ ranges from 1.15 to 1.74, $D_T$ ranges from 1.01 to 1.03, $d_f$ ranges from 1.64 to 1.82, and $D_f$ ranges from 2.69 to 2.83. The results (Figure 5f) also suggest that $\tau_{av}$ and the ratio $r_{\max}/r_{3D,\max}$ are linearly dependent. When $\tau_{av}$ is assigned as unity, the ratio $r_{\max}/r_{3D,\max}$ is estimated to be 0.93, which is approximately equal to unity. Similar findings are shown in Figure 4e.

**Dataset of Berryman and Blair [31]:** Based on digital image analysis, Berryman and Blair first studied the two-point correlation functions, porosity, and specific surface area of glass beads (55 µm), Ironton Galesville (IG) sandstones, and Berea sandstones [31]. Then, they predicted the permeability of these samples with the following equation:

$$K_{3D} = \frac{\varphi_{2D}^2}{bFS_{2D}^2},$$ (17)

where $F$ is the formation factor, and $b$ is a constant that depends on the cross-section of the tubes. It is recommended that $b$ is assigned as 2 and 3 for circular tubes and flat cracks, respectively.

In order to further verify our derived model, the predictions from our derived model were compared against the available data of Berryman and Blair [31]. Taking sample 3 (sample Berea 1) with an image specific surface of 0.0281 µm$^{-1}$, an areal porosity $\varphi_{2D}$ of 17%, and an image permeability $K_{2D}$ of 0.312 µm$^2$. For example, Berryman and Blair found the bulk porosity (laboratory porosity) $\varphi_{3D}$ ranged from 15% to 18% with an average value of 16.5%. In addition, based on Equation (14), the laboratory permeability of $K_{3D}$ was determined to be 0.023 µm$^2$. In our modeling, $\varphi_{3D}$ and $\varphi_{2D}$ were assigned as 16.5% and 17%, respectively. The results (Figure 6a) suggest that the predicted results ($K_{2D}$ and $K_{3D}$) of these samples (i.e., two glass beads, four Berea samples, IG-775 and IG-785) provide a good match over the test data. In addition, the pore structure parameters (e.g., $d_f$, $\tau_{av}$, $r_{\max}/r_{3D,\max}$, $D_T$, and $D_f$) of each sample are presented in Figure 6b. As pore structures ($d_f$, $\tau_{av}$, $r_{\max}/r_{3D,\max}$, $D_T$, and $D_f$) of these samples are different, the curves show different behaviors. In addition, the results (Figure 6c) also suggest that $\tau_{av}$ is remarkably correlated linearly with the ratio $r_{\max}/r_{3D,\max}$.

For a given porous medium (bulk porosity $\varphi_{3D}$) composed of identical spherical grains, the average particle radius $R_{ap}$ of the porous medium could be written as [26,31,58]:

$$S_{3D} = 4\pi R_{ap}^2 / \left( \frac{4\pi R_{ap}^3/3}{1-\varphi_{3D}} \right) = \frac{3(1-\varphi_{3D})}{R_{ap}} \Rightarrow R_{ap} = \frac{3(1-\varphi_{3D})}{S_{3D}},$$ (18)

where $S_{3D}$ is the specific surface area. Based on the derivation in Appendix C, $R_{ap}$ can be also expressed as:

$$R_{ap} = \lambda R_{avs} + (1-\lambda)R_{avc},$$ (19)

where $\lambda$ is the weight coefficient of the average spherical particle radius, $R_{avs}$ is the average spherical particle radius, and $R_{avc}$ is the average circular particle radius. More details about $R_{avs}$ and $R_{avc}$ can be found in Equations (A22) and (A24). By combining Equations (15) and (16), the weight coefficient of the average spherical particle radius can be determined. Figure 6d compares the calculated average particle radius for spherical particles and the calculated average particle radius for circular particles with that from the former model. The results from Figure 6d demonstrate that the calculated average particle radius from Equation (15) is approximately in the middle between the calculated values from Equations (A22) and (A24). This indicates that the model gives predicted values that are quite consistent with the results from the former model. In addition, by combining the results from Equations (15) and (16), the weight coefficients for different samples have been determined in Figure 6d. Taking sample 3 for example, when the weight coefficient is unity, the average particle radius can be effectively determined by Equation (A22).

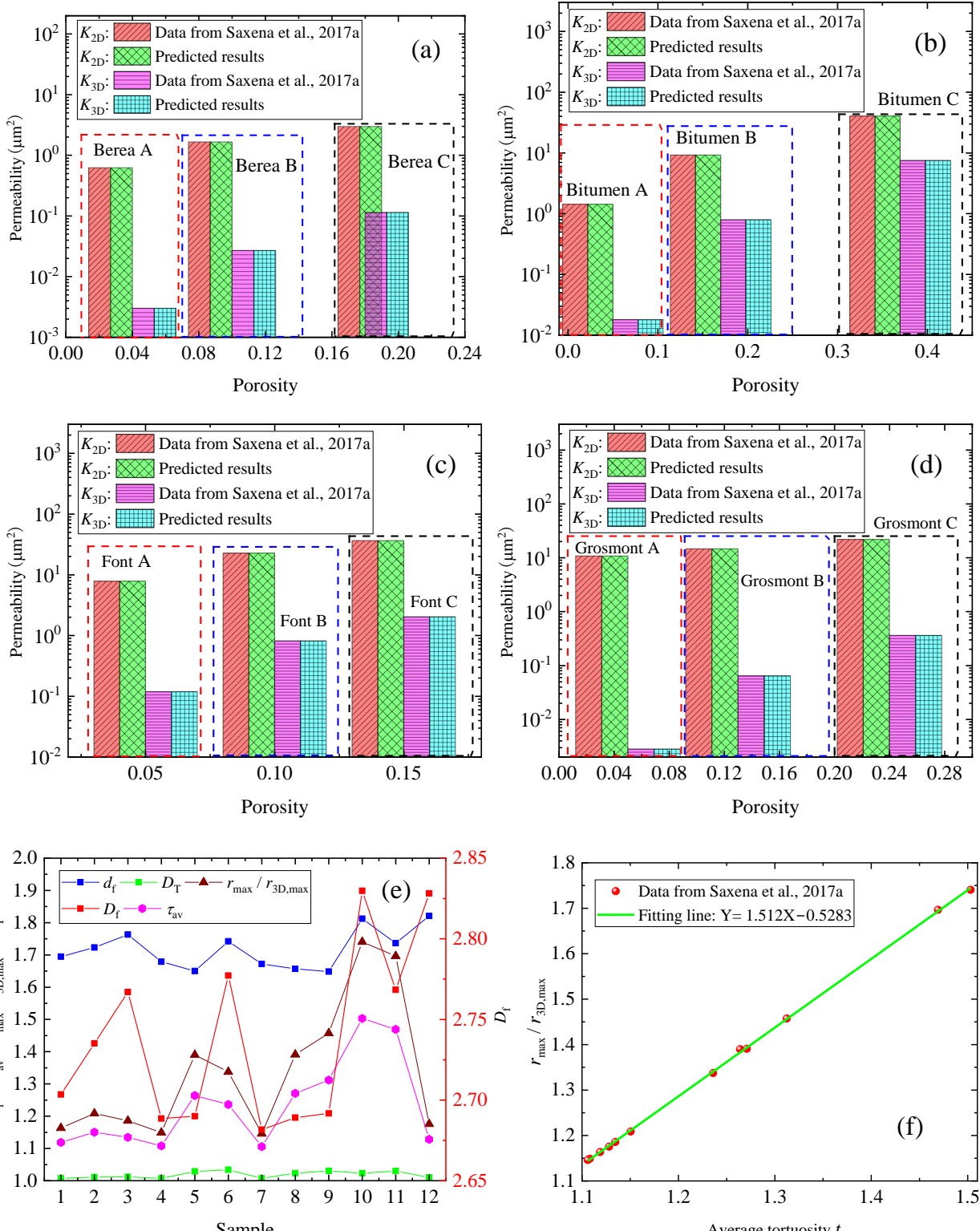

**Figure 5.** Modeling results of the samples [26]: (**a**) the predicted permeabilities from the derived model versus the data for Berea samples; (**b**) the predicted permeabilities from the derived model versus the data for Bitumen samples; (**c**) the predicted permeabilities from the derived model versus the data for Font samples; (**d**) the predicted permeabilities from the derived model versus the data for Grosmont samples; (**e**) the determined pore structure parameters for various samples; (**f**) $\tau_{av}$ versus the ratio $r_{max}/r_{3D,max}$ for various samples.

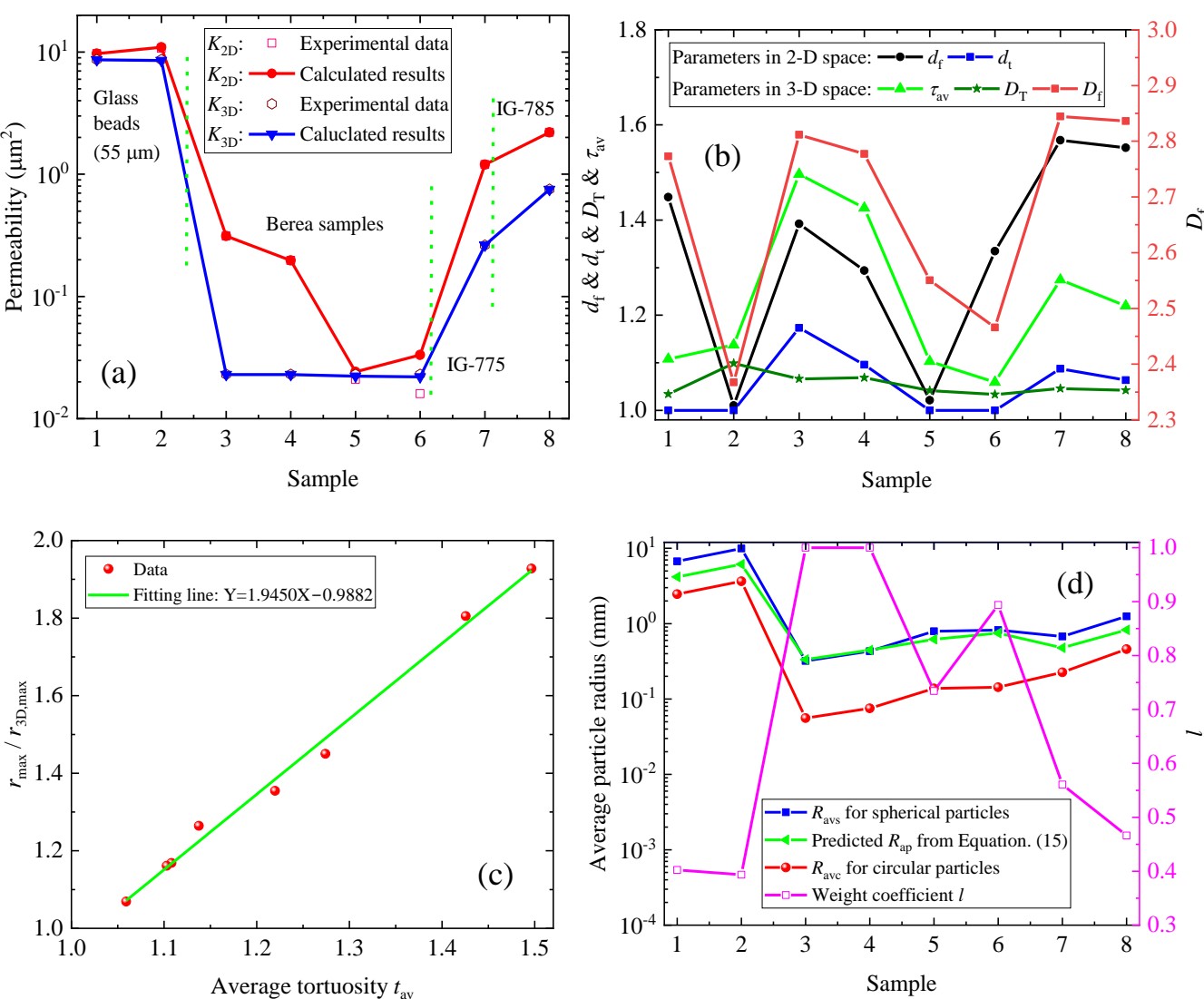

**Figure 6.** Modeling results of the samples: (**a**) predicted permeability ($K_{2D}$ and $K_{3D}$) and experimental data; (**b**) the determined pore structure parameters ($d_f$, $d_t$, $\tau_{av}$, $D_T$, and $D_f$) of various samples; (**c**) $\tau_{av}$ versus the ratio $r_{max}/r_{3D,max}$ for various samples; (**d**) the predicted average particle radius for spherical particles and circular particles versus that from Equation (15).

## 4. Results and Discussions

After validation with exhaustive experimental data, this derived model was utilized for sensitivity analysis of different parameters ($D_f$ and $D_T$) on the 3D permeability of porous media. Figure 7 shows the influence of pore fractal dimension $D_f$ on the average tortuosity $\tau_{av}$ and $K_{3D}$ of porous media in 3D. In this case, the maximum and minimum pore radii in 2D were 3 μm and 0.13 μm, respectively. The parameters $\varphi_{2D}$ and $\varphi_{3D}$ were both 0.15, and parameter $D_T$ was 1.1. During the calculation, the parameter $D_f$ ranged from 2.2 to 2.8. Based on our derived model, the maximum pore radius in 3D space $r_{3D,max}$ was determined, which ranged from 1.4 μm to 0.82 μm. In addition, the determined parameter $r_{3D,min}$ ranged from 0.0077 μm to 0.0045 μm. Specifically, when parameter $D_f$ was assigned as 2.2, the corresponding values of $r_{3D,max}$ and $r_{3D,min}$ were 1.4 μm and 0.0077 μm, respectively. As one can see from Figure 7a, there exists a positive relationship between $D_f$ and the average tortuosity $\tau_{av}$. The main reason is that a larger value of $D_f$ means a more complex pore structure of porous materials, leading to a larger value of $\tau_{av}$. Furthermore, the permeability of porous media in 3D decreases with an increase in $D_f$, which is anticipated. Figure 7b demonstrates that the linear correlativity between $\tau_{av}$

and the ratio $r_{max}/r_{3D,max}$ is very prominent. Moreover, when $\tau_{av}$ is assigned as unity, $r_{max}/r_{3D,max}$ is determined to be 0.9951, which is close to unity.

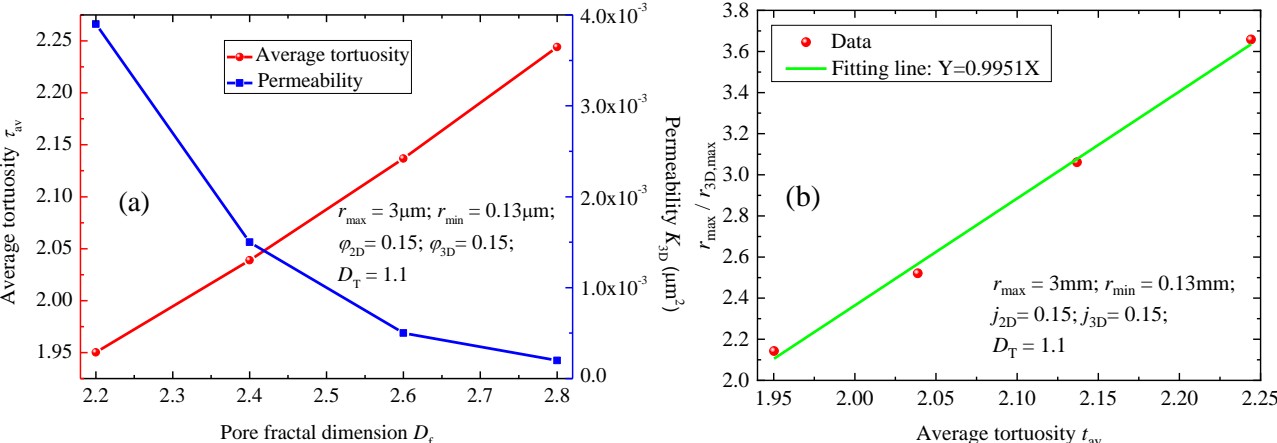

**Figure 7.** Effect of pore fractal dimension $D_f$ on properties of porous media in 3D: (**a**) curves of $D_f$ versus $\tau_{av}$ and $K_{3D}$; (**b**) $\tau_{av}$ versus the ratio $r_{max}/r_{3D,max}$.

The influence of parameter $D_T$ on the $\tau_{av}$ and $K_{3D}$ are shown in Figure 8. In this case, the basic parameters applied in the model are summarized in the corresponding figures. During the calculation, the parameter $D_f$ was assigned as 2.2, and the parameter $D_T$ ranged from 1.05 to 1.2. Based on our derived model, the parameter $r_{3D,max}$ ranges from 2.2 μm to 0.07 μm. The results (Figure 8a) suggest that a larger $\tau_{av}$ corresponds to a larger $D_T$. However, $K_{3D}$ decreases as $D_T$ increases. Specifically, for this case, when $D_T$ increases up to a certain value (e.g., $D_T \geq 1.2$), the value of $K_{3D}$ is extremely small. The main reason is that a larger value of $D_T$ means larger seepage resistance, resulting in a small value of $K_{3D}$. Figure 8b also reveals that $\tau_{av}$ has distinct linear correlations with the ratio $r_{max}/r_{3D,max}$. Similar findings have been also demonstrated in Figure 6.

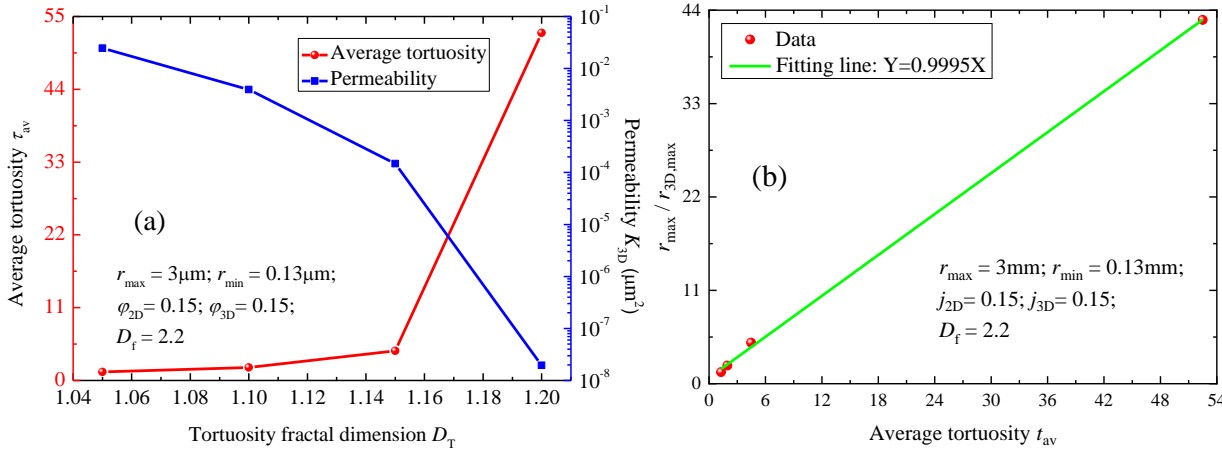

**Figure 8.** Effect of tortuosity fractal dimension $D_T$ on properties of porous media in 3D: (**a**) curves of $D_T$ versus $\tau_{av}$ and $K_{3D}$; (**b**) $\tau_{av}$ versus the ratio $r_{max}/r_{3D,max}$.

Based on the definition of the bridge function $f_{bridge}$ connecting $K_{2D}$ and $K_{3D}$ (i.e., $f_{bridge} = K_{2D}/K_{3D}$), we studied the influences of $D_f$ and $D_T$ on this bridge function $f_{bridge}$. Figure 9 presents the influences of $D_f$ and $D_T$ on $f_{bridge}$. For the calculation, the parameters applied in the model are summarized in the corresponding figures, which are identical to those for Figures 7 and 8, respectively. As one can see from Figure 9, the bridge function increases as $D_f$ (or $D_T$) increases. The main reason is that for a given 2D pore structure, a

larger value of $D_f$ (or $D_T$) corresponds to a smaller value of $K_{3D}$. Similar findings can be found in Figures 7a and 8a. Moreover, Figure 9 reveals that with an increase of $D_f$ (or $D_T$), the increase rate of $f_{bridge}$ increases. The main reason is that with an increase of $D_f$ (or $D_T$), the increase rate of the seepage resistance in the porous media increases.

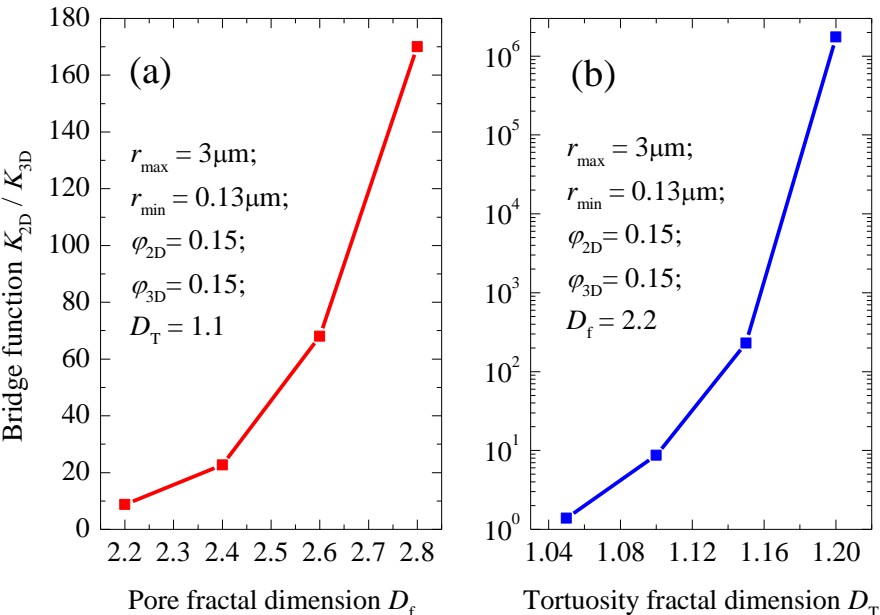

**Figure 9.** Effects of $D_f$ and $D_T$ on the bridge function $f_{bridge}$: (**a**) pore fractal dimension $D_f$ versus the bridge function $K_{2D}/K_{3D}$; (**b**) tortuosity fractal dimension $D_T$ versus the bridge function $K_{2D}/K_{3D}$.

**Model advantages and limitations:** The developed model provides a theoretical basis for predicting the 3D permeability of porous media from 2D digital image analysis without the need for 3D reconstruction, achieving high accuracy even with only 2D information. Compared to DRP, our model not only reduces the computational cost of high-resolution scanning, 3D pore network reconstruction, and numerical simulation but also captures the effect of realistic pore shapes on permeability. Furthermore, our model can be used for inverse modeling to estimate relevant parameters, such as tortuosity and the pore fractal dimension in 3D, making it highly practical for estimating permeability in heterogeneous porous media. Overall, our derived model is of great practical significance, as 2D images are easier and cheaper to access than 3D images and reconstructions of 3D pore networks.

However, it is worth noting that our derived model shares the problems of uncertainty and challenge in terms of predicting 3D pore structures and the permeability of porous media. Physically speaking, for a given 2D image, the possible 3D samples are infinite. That is, for a given 2D image, there are infinitely many permeabilities to this problem. As our model is derived based on fractal theory, it is limited to fractal porous media; however, it may not be suitable for some porous media. In addition, in our modeling, the areal porosity of 2D cross-sections $\varphi_{2D}$ is assumed to be identical to the bulk porosity of the 3D matrix. However, $\varphi_{2D}$ varies with the slices, and sometimes the difference between the maximum and the minimum 2D porosity is relatively large. Although $\varphi_{3D}$ is approximately in the middle between the maximum and minimum values of $\varphi_{2D}$, 2D porosity is not enough to represent $\varphi_{3D}$. Moreover, our developed model is limited to intact porous media and ignores the effect of micro-fractures on the 3D permeability of porous media. Thus, further research is required to reduce the uncertainty in estimating the 3D permeability of porous media from 2D images without reconstruction. Furthermore, in general, the pore surface of porous materials is rough, and the surface fractal dimension is crucial to characterize the fluid flow in porous media [41,59,60]. For example, Lei et al. [59] and Xiao et al. [41] derived theoretical models to study the fluid flow in porous media, and they concluded that the

effects of surface fractal dimension on permeability and relative permeability in porous media were significant. Thus, to make our model more reasonable, in our future work, a rough pore surface and surface fractal dimensions will be taken into account. Moreover, as mentioned above, the interaction between solid minerals and fluids will significantly affect the permeability of porous media. To improve the applicability of our derived model, the interaction between solid minerals and fluids will be taken into account. What is more, as the information on fracture systems may not be obtained from 2D cross-sections, in this paper, our derived model focuses on predicting the 3D permeability of intact porous media, and the fractures are ignored. In our future work, we will try to extend our model to study the 3D permeability of fractured porous media from 2D cross-sections of parent samples without 3D reconstruction.

## 5. Conclusions

In this paper, a novel analytical model was derived to estimate the 3D permeability of porous media from 2D cross-sectional images without reconstruction. Based on fractal theory and the Kozeny–Carman equation, a bridge function was developed to correlate the 2D pore information from the 2D images and the 3D pore structure of the parent samples. The derived model was validated with the results from lattice Boltzmann method (LBM) simulations and various experimental data and is shown to perform well.

The derived model was also used to conduct sensitivity analysis of different parameters (e.g., tortuosity fractal dimension in 3D space $D_T$, pore fractal dimension in 3D space $D_f$) on the 3D permeability $K_{3D}$ of porous media. The results indicate that the average tortuosity $\tau_{av}$ decreases with an increase of $D_f$ (or $D_T$). In addition, the average tortuosity $\tau_{av}$ is remarkably correlated linearly with the ratio of $r_{max}$ (the maximum pore radius of 2D images) to $r_{3D,max}$ (the maximum pore radius of 3D porous media). Moreover, the permeability $K_{3D}$ decreases as $D_f$ (or $D_T$) increases. With an increase of $D_f$ (or $D_T$), the decrease rate of the permeability $K_{3D}$ increases.

The proposed model not only reveals the Intrinsic link of 2D pore information and 3D pore structure in unconventional oil/gas reservoirs but also reduces the computational cost of high-resolution scanning and flow simulation to predict the 3D permeability of unconventional reservoirs. Our proposed model can be applied in many fields, such as $CO_2$ geology storage, unconventional onshore and offshore oil/gas development, and groundwater seepage.

Although the model focuses on predicting single-phase flow in dry porous media, when the irreducible wetting phase saturation is taken into account, our model can be extended to study the 3D effective permeability of porous media. In addition, it is also available to extend this model to study multi-phase flow in porous media and obtain relative permeability in porous media.

However, it should be noted that our proposed model never considers a rough pore surface (e.g., surface fractal dimension) or interactions in solid minerals–fluid systems. Thus, in our future work, more mechanisms will be taken into account to make our model more reasonable.

**Author Contributions:** Conceptualization, G.L. and T.L.; methodology, G.L.; software, G.L.; validation, G.L., T.L. and Q.L.; formal analysis, G.L.; investigation, Q.L. and X.H.; resources, Q.L. and X.H.; data curation, G.L.; writing—original draft preparation, G.L.; writing—review and editing, G.L., T.L., Q.L. and X.H.; visualization, G.L. and T.L.; supervision, Q.L. and X.H.; project administration, T.L. and Q.L.; funding acquisition, G.L. and T.L. All authors have read and agreed to the published version of the manuscript.

**Funding:** This study is jointly supported by the GuangDong Basic and Applied Basic Research Foundation (grant number 2022A1515110376), the Fundamental Research Funds for the Central Universities, China University of Geosciences (Wuhan) (grant number 107-G1323523046) and the National Natural Science Foundation of China (grant number 22208329).

**Institutional Review Board Statement:** Not applicable.

**Informed Consent Statement:** Not applicable.

**Data Availability Statement:** All data have been provided in the paper.

**Acknowledgments:** The authors are grateful for financial support from the GuangDong Basic and Applied Basic Research Foundation (No. 2022A1515110376), the Fundamental Research Funds for the Central Universities, China University of Geosciences (Wuhan) (No. 107-G1323523046), the National Natural Science Foundation of China (22208329), and the Independent Prospective Basic Project of State Key Laboratory of Offshore Oil Exploitation in 2023.

**Conflicts of Interest:** We declare that we have no conflict of interest.

## Nomenclature

Latin symbols

| | |
|---|---|
| $A$ | The cross-sectional area of the representative element ($\mu m^2$) |
| $A_{av}$ | The average pore area ($\mu m^2$) |
| $b$ | A constant that depends on the cross-section of the tubes (dimensionless) |
| $c$ | Lattice sound speed, which is determined by $\Delta x/\Delta t$ (m/s) |
| $d_e$ | The Euclidean dimension is 2 in 2D space (dimensionless) |
| $d_f$ | The pore fractal dimension in 2D space (dimensionless) |
| $d_t$ | The tortuosity fractal dimension in 2D space (dimensionless) |
| $D_f$ | The pore fractal dimension in 3D space (dimensionless) |
| $D_T$ | The tortuosity fractal dimension in 3D space (dimensionless) |
| $\mathbf{e}_\alpha$ | Lattice velocities (m/s) |
| $\mathbf{e}_y$ | The unit vector along the y-axis (m/s) |
| $F$ | The formation factor (dimensionless) |
| $f$ | The pore size distribution in 2D space (dimensionless) |
| $f_{3D}$ | The pore size distribution in 3D space (dimensionless) |
| $f_\alpha(\mathbf{y}, t)$ | The evolution of the density distribution function ($kg/m^3$) |
| $K_{2D}$ | Permeability of the 2D cross-section ($10^{-3}$ $\mu m^2$) |
| $K_{3D}$ | 3D permeability of porous media in flow equivalence ($10^{-3}$ $\mu m^2$) |
| $K_{ij}$ | Permeability in the $i$ direction when the flow is driven in the $j$ direction ($10^{-3}$ $\mu m^2$) |
| $L$ | The actual streamlined length of a tortuous capillary ($\mu m$) |
| $L_j$ | The size of the computational domain in the $j$ direction ($\mu m$) |
| $L_u$ | The edge length of the cubical unit cell ($\mu m$) |
| $N$ | The total number of pores in the representative element of a 2D cross-section (dimensionless) |
| $N_{3D}$ | The total number of pores in a representative elementary volume (dimensionless) |
| $N_{total}$ | The total lattice number (dimensionless) |
| $p$ | The transient pressure (Pa) |
| $p_{in}$ | The inlet transient pressure (Pa) |
| $p_{out}$ | The outlet transient pressure (Pa) |
| $r$ | The pore radius in 2D space ($\mu m$) |
| $r_{av}$ | The average pore radius in 2D space ($\mu m$) |
| $r_{3D}$ | The pore radius in 3D space ($\mu m$) |
| $r_{3D,av}$ | The average pore radius in 3D space ($\mu m$) |
| $r_{max}$ | The maximum pore radius in 2D space ($\mu m$) |
| $r_{min}$ | The minimum pore radius in 2D space ($\mu m$) |
| $r_{3D,max}$ | The maximum pore radius in 3D space ($\mu m$) |
| $r_{3D,min}$ | The minimum pore radius in 3D space ($\mu m$) |
| $R$ | The particle radius of the porous media ($\mu m$) |
| $R_{ap}$ | The average particle radius of the porous media ($\mu m$) |
| $R_{avc}$ | The average circular particle radius ($\mu m$) |
| $R_{avs}$ | The average spherical particle radius ($\mu m$) |
| $R_{max}$ | The maximum particle radius ($\mu m$) |
| $S_{2D}$ | The specific surface area of the 2D cross-section ($\mu m^{-1}$) |
| $S_{3D}$ | The specific surface area of the 3D cross-section ($\mu m^{-1}$) |
| $\mathbf{u}_j$ | The velocity at the void point $j$ (m/s) |
| $\mathbf{u}$ | The flow velocity (m/s) |

| | |
|---|---|
| $V$ | The pore volume of representative elementary volume ($\mu m^3$) |
| $V_{av}$ | The average pore volume ($\mu m^3$) |
| $V_u$ | The volume of the cubical unit cell ($\mu m^3$) |
| Greek symbols | |
| $\tau$ | Tortuosity (dimensionless) |
| $\tau(r_{3D})$ | Tortuosity of pore radius $r_{3D}$ (dimensionless) |
| $\tau_{av}$ | The average tortuosity (dimensionless) |
| $\tau_0$ | The relaxation time (dimensionless) |
| $\varphi_{2D}$ | The areal porosity of 2D cross-sections (dimensionless) |
| $\varphi_{3D}$ | The bulk porosity of a 3D matrix (dimensionless) |
| $\mu$ | The dynamic viscosity (mPa·s) |
| $\Delta y$ | The lattice distance or the voxel size ($\mu m$) |
| $\mathbf{y}$ | The grid location ($\mu m$) |
| $\Delta t$ | The time step (s) |
| $\mathbf{u}_i$ | The component in the $i$ direction of the volumetric average velocity (m/s) |
| $v$ | The kinematic viscosity ($m^2/s$) |
| $\rho$ | The fluid density in the D3Q19 model ($kg/m^3$) |
| $\omega_\alpha$ | Fixed weighting factors (dimensionless) |
| $\pi$ | Circular constant, which is approximately equal to 3.1415926 (dimensionless) |
| $\lambda$ | The weight coefficient of the average spherical particle radius (dimensionless) |
| Subscript symbols | |
| $\alpha$ | The direction in the D3Q19 model (dimensionless) |
| 2D | Two-dimensional space |
| 3D | Three-dimensional space |
| max | Maximum |
| min | Minimum |

## Appendix A. Fractal Theory of Porous Media in 2D and 3D Space

As mentioned above, for a 2D cross-section, pore structure parameters can be determined using digital image process technologies and thin section analysis. For example, pore fractal dimension $d_f$ can be calculated with the box-counting algorithm [38]. Moreover, with thin section analysis or digital image processing techniques, areal porosity $\varphi_{2D}$ and pore size distribution can be easily determined. As the pore structures of most sedimentary porous media follow fractal characteristics, fractal theory has been widely applied to describe the pore structures of porous media [29,34,35]. Based on fractal geometry, with the determined pore fractal dimension $d_f$, the maximum pore radius $r_{max}$, and the minimum pore radius $r_{min}$, the pore size distribution $f$ and the total number of pores in a representative element of 2D cross-section $N$ can be written as [29,35]:

$$\begin{cases} f = d_f r_{min}^{d_f} r^{-(d_f+1)}, \\ N = \left(\frac{r_{max}}{r_{min}}\right)^{d_f}. \end{cases} \tag{A1}$$

Based on Equation (A1), the average pore radius $r_{av}$ and specific surface area $S_{2D}$ of the 2D cross-section are:

$$\begin{cases} r_{av} = \int_{r_{min}}^{r_{max}} r f dr = \frac{d_f r_{min}}{d_f-1}\left[1 - \left(\frac{r_{min}}{r_{max}}\right)^{d_f-1}\right]; \\ S_{2D} = \frac{\varphi_{2D}}{A} N \int_{r_{min}}^{r_{max}} 2\pi r f dr = \frac{\varphi_{2D}}{A} 2\pi r_{av} \left(\frac{r_{max}}{r_{min}}\right)^{d_f}, \end{cases} \tag{A2}$$

where $A$ is the cross-sectional area of the representative element (RE), which is:

$$A = \frac{N}{\varphi_{2D}} \int_{r_{min}}^{r_{max}} \pi r^2 f dr = \frac{\pi d_f r_{max}^{d_f}\left(r_{max}^{2-d_f} - r_{min}^{2-d_f}\right)}{\varphi_{2D}(2-d_f)} = \frac{\pi d_f r_{max}^2}{\varphi_{2D}(2-d_f)}\left(1 - \frac{r_{min}^{2-d_f}}{r_{max}^{2-d_f}}\right). \tag{A3}$$

By combining the Hagen–Poiseulle equation, the bundle of the capillary tube model, and Darcy's law, the permeability of the 2D cross-section (i.e., the 2D composite permeated with tortuous capillary tubes that are perpendicular to the 2D cross-section, Figure 1) $K_{2D}$ can be obtained as [12,29,40,54,55]:

$$K_{2D} = \frac{\pi d_f r_{max}^{d_f} \left( r_{max}^{3-d_f+d_t} - r_{min}^{3-d_f+d_t} \right)}{2^{4-d_t} \sqrt{A^{1+d_t}} (3 - d_f + d_t)}, \tag{A4}$$

where $d_t$ is the tortuosity fractal dimension in 2D space, which can be determined with [42]:

$$d_t = (2 - d_f + 1) + (2 - d_f) \frac{\log(d_f) - \log(d_f - 1)}{\log(\varphi_{2D})}. \tag{A5}$$

when $d_t$ is assigned as unity, $K_{2D}$ can be simplified as:

$$K_{2D} = \frac{\pi d_f r_{max}^{d_f} \left( r_{max}^{4-d_f} - r_{min}^{4-d_f} \right)}{8A(4 - d_f)}. \tag{A6}$$

For 3D pore space, with the given pore fractal dimension $D_f$, the maximum pore radius $r_{3D,max}$, and the minimum pore radius $r_{3D,min}$, the pore size distribution $f_{3D}$ and the total number of pores in a representative elementary volume (SRV) $N_{3D}$ can be written as:

$$\begin{cases} f_{3D} = D_f r_{3D,min}^{D_f} r_{3D}^{-(D_f+1)}; \\ N_{3D} = \left( \dfrac{r_{3D,max}}{r_{3D,min}} \right)^{D_f}, \end{cases} \tag{A7}$$

where the subscript 3D presents 3D space. Based on fractal theory, in 3D pore space, the actual streamline length $L$ of a tortuous capillary and tortuosity fractal dimension $D_T$ can be determined as [29,42]:

$$\begin{cases} L = (2r_{3D})^{1-D_T} V^{\frac{D_T}{3}}; \\ D_T = (3 - D_f + 1) + (3 - D_f) \dfrac{\log(D_f) - \log(D_f - 1)}{\log(\varphi_{3D})}. \end{cases} \tag{A8}$$

where pore volume $V$ of a representative elementary volume (REV) can be obtained as [61]:

$$V = \frac{N_{3D}}{\varphi_{3D}} \int_{r_{3D,min}}^{r_{3D,max}} \pi r_{3D}^2 L dr_{3D}. \tag{A9}$$

By rewriting Equation (A9), we have:

$$V = \left\{ \frac{\pi D_f r_{3D,max}^{3-D_T}}{4\varphi_{3D}(3 - D_T - D_f)} \left[ 1 - \left( \frac{r_{3D,min}}{r_{3D,max}} \right)^{3-D_T-D_f} \right] \right\}^{\frac{3}{3-D_T}}. \tag{A10}$$

By combining the Hagen–Poiseulle equation, the bundle of the capillary tube model, and Darcy's law, the 3D permeability $K_{3D}$ can be obtained as [12,55,61,62]:

$$K_{3D} = \frac{N_{3D}}{V^{\frac{2}{3}}} \int_{r_{min}}^{r_{max}} \frac{\pi r_{3D}^4}{8(2r_{3D})^{1-D_T} V^{\frac{D_T-1}{3}}} f dr = \frac{\pi D_f 2^{D_T} r_{3D,max}^{D_f} \left( r_{3D,max}^{3-D_f+D_T} - r_{3D,min}^{3-D_f+D_T} \right)}{16 V^{\frac{1+D_T}{3}} (3 - D_f + D_T)}. \tag{A11}$$

By combining Equation (A7) and Equation (8), the average tortuosity $\tau_{av}$ is:

$$\tau_{av} = \int_{r_{3D,min}}^{r_{3D,max}} f_{3D} \left( \frac{\sqrt[3]{V}}{2r_{3D}} \right)^{D_T-1} dr_{3D} = 2^{1-D_T} D_f r_{3D,min}^{D_f} \left( \sqrt[3]{V} \right)^{D_T-1} \frac{r_{3D,max}^{1-D_f-D_T} - r_{3D,min}^{1-D_f-D_T}}{1-D_f-D_T}. \tag{A12}$$

Then, by the definition of specific surface area $S_{3D}$ in 3D space, we have:

$$S_{3D} = \frac{N_{3D}}{V} \int_{r_{3D,min}}^{r_{3D,max}} 2\pi r_{3D} \sqrt[3]{V^{D_T}} (2r_{3D})^{1-D_T} f_{3D} dr_{3D}$$
$$= 2^{2-D_T} \pi D_f V^{\frac{D_T-3}{3}} r_{3D,max}^{D_f} \frac{r_{3D,max}^{2-D_T-D_f} - r_{3D,min}^{2-D_T-D_f}}{2-D_T-D_f}. \tag{A13}$$

**Appendix B. The Algorithm of the D3Q19 Lattice Boltzmann Method**

For single-phase flow simulations, the lattice Boltzmann method (LBM) based on simple BGK relaxation was used here. The fundamental idea of the LBM is to construct a kinetic model simplified from the Boltzmann equation at mesoscale that incorporates the essential physical processes to ensure that the computed macroscopic quantities satisfy the desired governing equation (i.e., Navier–Stokes equation in the current study). The computational grids/lattices are uniformly distributed inside the computational domain. The only unknown $f_\alpha(x, t)$ is the density distribution function and evolves according to the following propagation–collision scheme [63–66]:

$$f_\alpha(\mathbf{y} + \Delta t \mathbf{e}_\alpha, \, t + \Delta t) = f_\alpha(\mathbf{y}, \, t) + \frac{f_\alpha^{eq}(\mathbf{y}, \, t) - f_\alpha(\mathbf{y}, \, t)}{\tau_0}, \tag{A14}$$

and

$$f_\alpha^{eq}(\mathbf{y}, \, t) = \rho \omega_\alpha \left[ 1 + \frac{3}{c^2} \mathbf{e}_\alpha \cdot \mathbf{u} + \frac{9}{2c^4} (\mathbf{e}_\alpha \cdot \mathbf{u})^2 - \frac{3}{2c^2} \mathbf{u} \cdot \mathbf{u} \right], \tag{A15}$$

where $\Delta t$ is the timestep, $\mathbf{y}$ is the grid location, $c = \Delta y / \Delta t$ is computed with the grid distance $\Delta y$ (namely the voxel size here), and $\tau_0$ is the dimensionless relaxation time, which is determined by the kinematic viscosity $v$, namely $\tau_0 = 3v/(c\Delta y) + 0.5$. Additionally, the lattice velocity $\mathbf{e}_\alpha$ and weight $\omega_\alpha$ in the moving direction $\alpha \in [0, Q-1]$ were selected according to the D3Q19 model (3-dimension and 19-velocity) [64]. For an arbitrary grid at $\mathbf{y}$, its neighboring grids are exactly located at $\mathbf{y} + \mathbf{e}_\alpha \Delta t$ since the magnitude of $\mathbf{e}_\alpha$ depends on $c$. The equilibrium distribution function $f_\alpha^{eq}$ was determined from the local fluid density $\rho$ and flow velocity $\mathbf{u}$ that were computed in the absence of internal/external force as follows:

$$\begin{cases} \rho = \sum\limits_{\alpha=0}^{18} f_\alpha; \\ \mathbf{u} = \frac{1}{\rho} \sum\limits_{\alpha=0}^{18} \mathbf{e}_\alpha f_\alpha. \end{cases} \tag{A16}$$

At the initial state, $f_\alpha$ is equal to $f_\alpha^{eq}$, which was computed with the initial distributions of $\rho$ and $\mathbf{u}$. The bounce-back scheme was used at the complicated solid surface for the non-slip boundary condition and was naturally implemented as the half-way bounce-back scheme with a higher accuracy because the actual solid–fluid interface was exactly located in the middle between the solid and fluid grids in the digital rock simulations. The flows in the current study were driven by a pressure difference $p_{in} - p_{out}$ between the inlet and outlet (equivalent to a density difference due to $p = \rho c^2/3$), and the density constraint was imposed by the robust non-equilibrium extrapolation scheme [67]. The permeability was computed after convergence:

$$K_{ij} = \frac{\rho v L_j}{p_{in} - p_{out}} \langle \mathbf{u}_i \rangle = \frac{\rho v L_j}{p_{in} - p_{out}} \frac{1}{N_{total}} \sum_{j \in void} \mathbf{u}_j, \tag{A17}$$

where $\langle \mathbf{u}_i \rangle$ is the component in the $i$ direction of the volumetric average velocity and computed by a summation of $\mathbf{u}_i$ over the void/fluid grids, $N_{\text{total}}$ is the total number of both void and solid grids, $K_{ij}$ is the component in the $i$ direction when the flow is driven in the $j$ direction, and $L_j$ is the size of the computational domain in the $j$ direction. The permeability $K_{ij}$ as a tensor has nine components in total by driving the flow in three different directions using three independent simulations. However, the current simulations in this paper always drive the flow by a pressure difference along the $j = y$ direction and focus on the component in the $i = y$ direction, and $K_{yy}$ is denoted by $K_{3D}$ as the 3D permeability of the porous media.

It is worth noting that attention has been paid to alleviating the velocity and permeability dependences on the relaxation time due to the nonzero Knudsen effect in the LBM simulations [46,68] where the Knudsen (Kn) number is $\sqrt{\pi/6}(\tau_0 - 0.5)/N_{\text{pore}}$ [69], and $N_{\text{pore}}$ is the grid number used to discretize the dominant pore size. However, we could not set $(\tau_0 - 0.5) \to 0$ for $Kn \to 0$; that would result in zero kinematic viscosity and an infinite Reynold (Re) number, leading to the dependence of the permeability on Re. In practical simulations of digital rocks [46,68], using $(\tau_0 - 0.5)$ around 0.2 and a small relative pressure drop of only 1% (i.e., $p_{\text{in}} - p_{\text{out}} = 0.01p_{\text{in}}$) can make the Kn and Re effects negligible, respectively. On the other hand, Li et al. (2018) found the multi-relaxation time (MRT) model might provide a parameter range that was wider than that of the current BGK model, but the basic mechanism remained the same; i.e., in simulations of intrinsic permeability, one should choose the appropriate relaxation parameter to ensure that Kn is small (but also nonzero as in the current BGK model).

**Appendix C. Determination of the Average Particle Size**

In this section, the average rock particle radius in porous media with bulk porosity $\varphi_{3D}$ was estimated with the idealized geometrical model shown in Figure A1, assuming that the unit cell was composed of spherical particles of the same size. Based on Figure A1, the volume of the cubical unit cell $V_u$ and its edge length $L_u$ are:

$$\begin{cases} V_u = \frac{4\pi R^3}{3(1-\varphi_{3D})}; \\ L_u = \sqrt[3]{\frac{4\pi R^3}{3(1-\varphi_{3D})}} = \sqrt[3]{\frac{4\pi}{3(1-\varphi_{3D})}} R. \end{cases} \tag{A18}$$

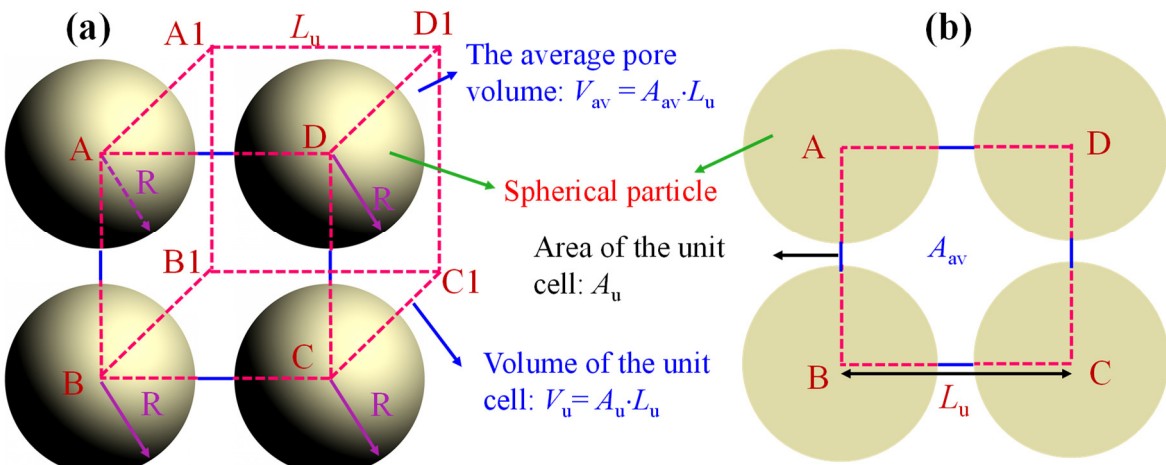

**Figure A1.** The arrangement of spherical rock particles for the average pore radius of porous media: (**a**) the cubical unit cell; (**b**) front view of the unit cell.

Based on Equation (A18), the average pore volume $V_{av}$ is:

$$V_{av} = V_u - \frac{4\pi R^3}{3} = \frac{4\pi R^3 \varphi_{3D}}{3(1 - \varphi_{3D})}. \tag{A19}$$

Mathematically, the average pore volume $V_{av}$ can be also estimated with:

$$V_{av} = A_{av} L_u = \pi r_{3D,av}^2 \sqrt[3]{\frac{4\pi}{3(1 - \varphi_{3D})}} R, \tag{A20}$$

wherein $r_{3D,av}$ is the average pore radius in 3D space, which is:

$$r_{3D,av} = \int_{r_{3D,min}}^{r_{3D,max}} r_{3D} f_{3D} dr_{3D} = \frac{D_f r_{3D,min}}{D_f - 1} \left[ 1 - \left( \frac{r_{3D,min}}{r_{3D,max}} \right)^{D_f - 1} \right]. \tag{A21}$$

By combining Equations (A19) and (A20), the average spherical particle radius $R_{avs}$ is

$$R_{avs} = r_{3D,av} \sqrt{\sqrt[3]{\frac{4\pi}{3(1 - \varphi_{3D})}} \frac{3(1 - \varphi_{3D})}{4\varphi_{3D}}}. \tag{A22}$$

By simplifying spherical particles in Figure A1 as circular particles, Xu and Yu derived an analytical model for the maximum particle radius $R_{max}$, which is [12]:

$$R_{max} = r_{3D,max} \sqrt{\frac{\varphi_{3D}}{1 - \varphi_{3D}}}. \tag{A23}$$

In light of Equation (A23), the average circular particle radius $R_{avc}$ can be also determined as:

$$R_{avc} = r_{3D,av} \sqrt{\frac{\varphi_{3D}}{1 - \varphi_{3D}}}. \tag{A24}$$

Then, by using the average particle radius statistical average method, the average particle radius $R_{ap}$ of the porous media can be expressed as:

$$R_{ap} = \lambda R_{avs} + (1 - \lambda) R_{avc}, \tag{A25}$$

wherein $\lambda$ is the weight coefficient of the average spherical particle radius.

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
