# Peer review of "Estimating Permeability of Porous Media from 2D Digital Images"

_jmse, doi:10.3390/jmse11081614_

Round 1

Reviewer 1 Report

In this study, the authors introduce a new method that combines the KC equation and fractal theory to establish a bridge between 2D cross-sectional images and 3D pore structures of parent samples, in order to predict permeability. They validate their model through Lattice Boltzmann simulations on carbonate samples and comparison with existing data. Results show accurate permeability estimates using 2D images, simplifying determination in heterogeneous porous media and revealing correlations between 2D and 3D properties. I find this work interesting, within the aim and scope of JMSE and would be happy to recommend its publication after these comments are addressed:

1. The Introduction reads well; however, one key part is missing, which is a brief discussion on the application of lattice Boltzmann as a promising technique for the prediction of pore-scale properties including permeability, relative permeability, and so on. I suggest the authors add a paragraph to the introduction, they can refer to several works conducted with LBM including these two recently published studies:

-       Vasheghani Farahani and Mousavi Nezhad, 2022. On the effect of flow regime and pore structure on the flow signatures in porous media. Physics of Fluids, 34(11).

-       Soleimani et al., 2022. Analysis of Marangoni Effects on the Non-isothermal Immiscible Rayleigh-Taylor Instability. International Journal of Multiphase Flow, 156, p.104231.

-       Norouzi et al., 2019, June. Pore-scale simulation of capillary force effect in water-oil immiscible displacement process in porous media. In 81st EAGE Conference and Exhibition 2019 (Vol. 2019, No. 1, pp. 1-5). European Association of Geoscientists & Engineers.

2. In Eqs. (1-5), can the author add some details about how r_max and r_min are determined in an image?

3. Line 216, Eq.(7a) should be corrected to Eq.(7), I think.

4. Again, one key missing part is LBM details; the information in Appendix B is not adequate. I believe it is necessary for the authors to provide their LBM details (model, boundary conditions, the collision operation, …). Mesh dependence analysis is also important to be detailed.

5. I see that the authors used a BGK collision operator for their LB simulations. While being simple and easy to implement, using the BGK collision operator may result in unphysical dependence of the dimensionless velocity field and permeability to the relaxation time. How did the authors address this issue?

6. Is this method able to predict the permeability of fractured porous structures?

7. A general concern is that the captions provided for Figures and Tables are inadequate. Please enrich them with more information regarding what is presented in Figures and Tables.

8. Can the authors comment on whether this model can be extended for predicting the effective permeability?

Author Response

In this study, the authors introduce a new method that combines the KC equation and fractal theory to establish a bridge between 2D cross-sectional images and 3D pore structures of parent samples, in order to predict permeability. They validate their model through Lattice Boltzmann simulations on carbonate samples and comparison with existing data. Results show accurate permeability estimates using 2D images, simplifying determination in heterogeneous porous media and revealing correlations between 2D and 3D properties. I find this work interesting, within the aim and scope of JMSE and would be happy to recommend its publication after these comments are addressed:

Response: Special thanks for your good comments. We have made the change as per your valuable suggests and comments, which has been colored in Red in the revised manuscript.

  1. The Introduction reads well; however, one key part is missing, which is a brief discussion on the application of lattice Boltzmann as a promising technique for the prediction of pore-scale properties including permeability, relative permeability, and so on. I suggest the authors add a paragraph to the introduction, they can refer to several works conducted with LBM including these two recently published studies:

Vasheghani Farahani and Mousavi Nezhad, 2022. On the effect of flow regime and pore structure on the flow signatures in porous media. Physics of Fluids, 34(11).

Soleimani et al., 2022. Analysis of Marangoni Effects on the Non-isothermal Immiscible Rayleigh-Taylor Instability. International Journal of Multiphase Flow, 156, p.104231.

Norouzi et al., 2019, June. Pore-scale simulation of capillary force effect in water-oil immiscible displacement process in porous media. In 81st EAGE Conference and Exhibition 2019 (Vol. 2019, No. 1, pp. 1-5). European Association of Geoscientists & Engineers.

Response: Special thanks for your good comments. We have made the change as per your valuable suggests and comments, which is as follows: “Recently, as a promising technique for modeling fluid flow in porous media at pore-scale, Lattice-Boltzmann Method (LBM) has been widely applied by scholars to predict properties (e.g., permeability, and relative permeability) of porous media (Norouzi et al., 2019; Khodja et al., 2020; Soleimani et al., 2022; Vasheghani and Mousavi, 2022; Lei et al., 2023b). For example, Khodja et al. (2020) stated that compared with the traditional methods of Computational Fluid Dynamics, LBM was ideally suited for massively parallel computation, and it had the advantages of simplicity and flexibility in dealing with complicated geometry. Thus, in this paper, D3Q19 LBM model will be applied to validate our derived model.” The changes are colored in red.

  • Norouzi S., Soleimani R., Farahani M.V., et al. Pore-scale simulation of capillary force effect in water-oil immiscible displacement process in porous media. Presented in the 81st EAGE Conference and Exhibition 2019.
  • Soleimani R., Azaiez J., Zargartalebi M., et al. Analysis of marangoni effects on the Non-isothermal immiscible Rayleigh-Taylor instability. International Journal of Multiphase Flow, 2022, 156: 104231.
  • Vasheghani F.M., Mousavi N.M. On the effect of flow regime and pore structure on the flow signatures in porous media. Physics of Fluids, 2022, 34(11): 115139.
  • Lei G., Xue L., Liao Q., et al. A novel analytical model for porosity-permeability relations of argillaceous porous media under stress conditions. Geoenergy Science and Engineering, 2023b, 225: 211659.

  1. In Eqs. (1-5), can the author add some details about how r_max and r_min are determined in an image?

Response: Special thanks for your good comments. We have made the change as per your valuable suggests and comments, which is as follows: “Moreover, by using digital image processing techniques, various pore structure parameters (pore size distribution, the maximum pore size, the minimum pore size, average pore radius, pore perimeter, and specific surface area) of 2D cross-sectional images can be easily determined (Blunt et al., 2013; Saxena et al., 2017b; Wu et al., 2019a; 2019b; Khodja et al., 2020). For example, for a given 2D digital image, parameters (rmax, rmin and φ2D) can be determined by using the Avizo software, which can effectively determine pore space morphology and extract pore networks (Saxena and Mavko, 2016).” The changes are colored in red.

  1. Line 216, Eq.(7a) should be corrected to Eq.(7), I think.

Response: Special thanks for your good comments. We have made the change as per your valuable suggests and comments. The changes are colored in red.

  1. Again, one key missing part is LBM details; the information in Appendix B is not adequate. I believe it is necessary for the authors to provide their LBM details (model, boundary conditions, the collision operation, …). Mesh dependence analysis is also important to be detailed.

Response: Special thanks for your good comments. We have improved the model description and provided all details in the revised manuscript, which is as follows:

“For single-phase flow simulations, the lattice Boltzmann method (LBM) based on the simple BGK relaxation is used here. The fundamental idea of the LBM is to construct a kinetic model simplified from the Boltzmann equation at mesoscale that incorporates the essential physical processes to ensure that the computed macroscopic quantities satisfy the desired governing equation (i.e., Navier–Stokes equation in the current study). The computational grids/lattices are uniformly distributed inside the computational domain. The only unknown  is the density distribution function and evolves according to the following propagation-collision scheme (Qian, et al., 1992; Sukop and Thorne, 2006; Li, 2020a): 

           (B1)

and

      (B2)

where Δt is the timestep, y is the grid location, c = Δy / Δt is computed by the grid distance Δy (namely the voxel size here), τ0 is the dimensionless relaxation time, which is determined by the kinematic viscosity v, namely . Additionally, the lattice velocity eα and weight ωα in the moving direction  are selected according to the D3Q19 model (3-dimension and 19-velocity) (Qian, et al., 1992). For arbitrary grid at y, its neighbouring grids are exactly located at y + eαΔt, since the magnitude of eα depends on c. The equilibrium distribution function  is determined from the local fluid density  and flow velocity u that are computed in the absence of internal/external force as follows:

                                      (B3)

At the initial state,  is equal to  that is computed by the initial distributions of  and u. The bounce back scheme is used at the complicated solid surface for the non-slip boundary condition and is naturally implemented as the half-way bounce back scheme with higher accuracy because the actual solid-fluid interface is exactly located at the middle between solid and fluid grids in digital rock simulations. The flows in the current study are driven by a pressure difference  between the inlet and outlet (equivalent to a density difference due to ) and the density constraint is imposed by the robust non-equilibrium extrapolation scheme (Guo et al., 2002). The permeability is computed after convergence:

          (B4)

where  is the component in the  direction of the volumetric average velocity and computed by a summation of  over the void/fluid grids,  is the total number of both void and solid grids,  is the component in the  direction when the flow is driven in the  direction,  is the size of computational domain in the  direction. The permeability  as a tensor has 9 components in total by driving the flow in three different directions using three independent simulations. However, the current simulations in this paper always drive the flow by a pressure difference along the  direction and focus on the component in the  direction, and  is denoted by  as the 3D permeability of porous media.

As the reviewer suggested, mesh dependence analysis is important in numerical simulation. And mesh dependence study is usually included in benchmark problems but not adopted in most digital rock studies, where the spatial grid/voxel number is already very large and it is usually not affordable to refine the grid by even just a factor of 2 (8 times in 3D). Additionally, the digital rock itself is a very rough approximation of the real rock by using only 3~5 voxels to discretize the dominant pore size. Therefore, even if we use much more computational cost to refine the simulation grids based on a rough digital rock, the simulation results are not necessary to be more accurate in comparison with the real rock properties. On the other hand, without using grid refinement, we can simulate a much larger digital rock to cover a larger physical domain of the real heterogeneous rock, which could strike a good balance between the local accuracy and global characterization.

  1. I see that the authors used a BGK collision operator for their LB simulations. While being simple and easy to implement, using the BGK collision operator may result in unphysical dependence of the dimensionless velocity field and permeability to the relaxation time. How did the authors address this issue?

Response: Special thanks for your good comments. Clarifications are added to the revised manuscript, which is as follows: “It is worth noting that attentions have been paid to alleviate the velocity and permeability dependences on the relaxation time due to the nonzero Knudsen effect in the LBM simulations (Li et al., 2018; Khodja et al., 2020), where the Knudsen (Kn) number is  (Zhang et al., 2005) and  is the grid number used to discretize the dominant pore size. However, we cannot set  for  that will result in zero kinematic viscosity and infinite Reynold (Re) number leading to the dependence of permeability on Re. In practical simulations of digital rocks (Li et al., 2018; Khodja et al., 2020), using  around 0.2 and a small relative pressure drop of only 1% (i.e., ) can make the Kn and Re effects negligible, respectively. On the other hand, Li et al. (2018) found the multi-relaxation-time (MRT) model might provide a parameter range that was wider than that of the current BGK model but the basic mechanism remained the same, i.e. in simulations of intrinsic permeability, one should choose appropriate relaxation parameter to ensure that Kn is small (but also nonzero as in the current BGK model).” The change is colored in red.

  • Zhang Y., Qin R., Emerson D.R. Lattice Boltzmann simulation of rarefied gas flows in microchannels. Physical Review E, 2005, 71(4): 047702.
  • Li J., Ho M.T., Wu L., et al. On the unintentional rarefaction effect in LBM modeling of intrinsic permeability. Advances in Geo-Energy Research, 2018, 2(4): 404-409.

  1. Is this method able to predict the permeability of fractured porous structures?

Response: Special thanks for your good comments. As stated in scholars (Zhao et al., 2018; Meng et al., 2020), LBM is able to predict the permeability of fractured porous structures. However, as the information of fracture systems may not be obtained from 2D cross-sections, in this paper, our derived model focuses on predicting 3D permeability of intact porous media, and the fractures are ignored. In our future work, we will try to extend our model to study the 3D permeability of fractured porous media from 2D cross-sections of parent samples without 3D reconstruction. Clarifications are added to the revised manuscript, which is as follows: “What’s more, as the information of fracture systems may not be obtained from 2D cross-sections, in this paper, our derived model focuses on predicting 3D permeability of intact porous media, and the fractures are ignored. In our future work, we will try to extend our model to study the 3D permeability of fractured porous media from 2D cross-sections of parent samples without 3D reconstruction.” The changes are colored in red.

  • Zhao Y., Wang Z., Ye J., et al. Lattice Boltzmann simulation of gas flow and permeability prediction in coal fracture networks. Journal of Natural Gas Science and Engineering, 2018, 53: 153-162.
  • Meng X., Wang L., Zhao W., et al. Simulating flow in porous media using the lattice Boltzmann method: Intercomparison of single-node boundary schemes from benchmarking to application. Advances in Water Resources, 2020, 141: 103583.

  1. A general concern is that the captions provided for Figures and Tables are inadequate. Please enrich them with more information regarding what is presented in Figures and Tables.

Response: Special thanks for your good comments. We have enriched the captions for Figures and Tables. The changes are colored in red.

  1. Can the authors comment on whether this model can be extended for predicting the effective permeability?

Response: Special thanks for your good comments. In essential, the derived model in this paper can be extended for predicting the effective permeability and relative permeability in porous media. In this paper, the model focuses on predicting single phase flow in dry porous media. In our future work, we will take irreducible wetting phase saturation into account to study the 3D effective permeability of porous media. In addition, it is also available to extend this model to study multi-phase flow in porous media and obtain relative permeability in porous media. Clarifications are added to the Conclusion, which is as follows: “Although the model focuses on predicting single phase flow in dry porous media, when the irreducible wetting phase saturation is taken into account, our model can be extended to study the 3D effective permeability of porous media. In addition, it is also available to extend this model to study multi-phase flow in porous media and obtain relative permeability in porous media.” The changes are colored in red.

Reviewer 2 Report

This paper show a novelty way to estimate permeability in a solid from image analysis with a theoretical model.

Please update references in section 1.

You write about fractal dimension, which one? box-counting? in the introduction, include a brief paragraph for kinds of fractal dimension, relationship with tortuosity and why you are using a specific one.

Please explain computer and characteristics for algorithm

Consider other properties in state-of-the-art. i.e. Suarez-Dominguez, et al. (2020). Mesoscopic model for the surface fractal dimension estimation of solid-solid and gas-solid dispersed systems. Surfaces and Interfaces, 18, 100407.

In part two please write the properties of the carbonate used in model validation.

In part 3 It is necessary a deeper explanation of how chemical nature of the solid is taken into account and how different carbonate media can possibly change results. Mention it also in limitation

In lines 374-386 explain if possible, why you are thinking the differences in results. Figure 6D has a similar behavior but 6b present a change in sample 6. why?

I strongly recommend include and appendix with symbols, variables and units used in the paper.

Conclusion needs to consider comments wrote here.

Please explain what is necessary in further work.

Author Response

  1. Please update references in section 1.

Response: Special thanks for your good comments. We have update references in Introduction (section 1) as per your valuable comments and suggestions. The changes are colored in Red in the revised manuscript.

  1. You write about fractal dimension, which one? box-counting? in the introduction, include a brief paragraph for kinds of fractal dimension, relationship with tortuosity and why you are using a specific one.

Response: Special thanks for your good comments. Pore fractal dimension and tortuosity fractal dimension are two key parameters characterizing fluid flow in porous media, thus, these two parameters are used in this work to model permeability of porous media. Mathematically speaking, pore fractal dimension can be obtained by using the box-counting algorithm (Ghiasi-Freez et al., 2012) and the theoretical models (Yu and Li, 2001; Yu and Cheng, 2002), and tortuosity fractal dimension can be predicted by using the model derived by Wei et al. (2015). We have added corresponding content in Introduction, which is as follows: “Previous studies reveal that interspaces in porous media possess fractal characteristics (Mandelbrot, 1982; Katz and Thompson, 1985; Yu and Li, 2001; Yu and Cheng, 2002; Xu and Yu, 2008; Wu et al., 2019b; Lei et al., 2021). Specifically, many scholars (Yu and Li, 2001; Yu and Cheng, 2002; Xu and Yu, 2008; Lei et al., 2021) suggested that pore fractal dimension and tortuosity fractal dimension were two key parameters characterizing fluid flow in porous media. Thus, in this paper, these two parameters will be applied to characterize permeability of porous media. As the box-counting algorithm (Ghiasi-Freez et al., 2012) and the theoretical models (Yu and Li, 2001; Yu and Cheng, 2002) can effectively predict pore fractal dimension (Xu and Yu, 2008; Cai et al., 2012; Lei et al., 2019; Xiao et al., 2022), in this paper, pore fractal dimension will be estimated by the box-counting algorithm and the theoretical models. In 2015, Wei et al. derived a tortuosity fractal dimension model for fractal porous media, which is related to porosity and pore fractal dimension. As this model is effective to determine tortuosity fractal dimension (Liu et al., 2021; Lei et al., 2023a), this model will be also used in this paper to model tortuosity fractal dimension.” And the change is colored in Red.

  • Liu L., Sun Q., Wu N., et al. Fractal analyses of the shape factor in Kozeny-Carman equation for hydraulic permeability in hydrate-bearing sediments. Fractals, 2021, 29(7): 2150217.
  • Lei G., Qu J., Wu Q., et al. Theoretical analysis of threshold pressure in tight porous media under stress. Physics of Fluids, 2023a, 35(7): 073313.
  • Xiao B., Li Y., Long G., et al. Fractal permeability model for power-law fluids in fractured porous media with rough surfaces. Fractals, 2022, 30(06): 2250115.

  1. Please explain computer and characteristics for algorithm

Response: Special thanks for your good comments. Based on our derived model, the computer and characteristics for algorithm can be summarized as follows:

Step 1: Determine 2D pore structure parameters (e.g., rmax, rmin, φ2D, and df) from 2D digital image. Specifically, for a given 2D digital image, parameters (rmax, rmin and φ2D) can be determined by using the Avizo software (Saxena and Mavko, 2016). In addition, df can be determined by box-counting algorithm (Ghiasi-Freez et al., 2012) or by Eq. (5). Subsequently, parameter φ3D can be calculated using Eq. (6).

Step 2: Determine parameter DT by solving Eq. (14). Then, determine Df by using Eq. (13). In addition, determine parameter V by using Eq. (15). Then, r3D,max and r3D,min can be determined by using Eq. (10).

Step 3: Determine K3D and τav of the corresponding parent sample using Eqs. (16) and (A12). In addition, the bridge function (i.e., the ratio of K2D to K3D) can be determined by combining Eqs. (1) and (16). Moreover, other 3D pore structure parameters of the parent sample in flow equivalence can be further estimated.

  1. Consider other properties in state-of-the-art. i.e. Suarez-Dominguez, et al. (2020). Mesoscopic model for the surface fractal dimension estimation of solid-solid and gas-solid dispersed systems. Surfaces and Interfaces, 18, 100407.

Response: Special thanks for your good comments. This paper focuses on predicting 3D permeability from 2D digital images without 3D reconstruction, to simplify the model, pore surface is assumed to be smooth in our manuscript. However, as the reviewer suggested, surface fractal dimension is a critical parameter to characterize fluid flow in porous media. In our future work, we will take rough pore surface and surface fractal dimension into account. We have added corresponding content in Model advantages and limitations, which is as follows: “Furthermore, in general, pore surface of porous materials is rough and surface fractal dimension is crucial to characterize fluid flow in porous media (Lei et al., 2018; Suarez-Dominguez et al. 2020; Xiao et al., 2022). For example, Lei et al. (2018) and Xiao et al. (2022) derived theoretical models to study fluid flow in porous media, and they concluded that the effects of surface fractal dimension on permeability and relative permeability in porous media was significant. Thus, to make our model more reasonable, in our future work, rough pore surface and surface fractal dimension will be taken into account.” And the change is colored in Red.

  • Lei G., Wang C., Wu Z., et al. Theory study of gas-water relative permeability in roughened fractures. Proceedings of the Institution of Mechanical Engineers, Part C: Journal of Mechanical Engineering Science, 2018, 232(24): 4615-4625.
  • Suarez-Dominguez E.J., Perez-Rivao A., Sanchez-Medrano M.T., et al. Mesoscopic model for the surface fractal dimension estimation of solid-solid and gas-solid dispersed systems. Surfaces and Interfaces, 2020, 18: 100407.
  • Xiao B., Li Y., Long G., et al. Fractal permeability model for power-law fluids in fractured porous media with rough surfaces. Fractals, 2022, 30(06): 2250115.

  1. In part two please write the properties of the carbonate used in model validation.

Response: Special thanks for your good comments. In model validation, firstly the 2D pore structure parameters (e.g., rmax, rmin, φ2D, and df) from 2D digital image will be determined, then, based on our derived model, the 3D permeability of carbonates will be predicted. In addition, our predicted 3D permeability is compared with that from D3Q19 LBM to validate the feasibility and effectiveness of our derived model. Thus, the properties of the carbonate used in model validation are the maximum pore radius, the minimum pore radius, porosity and pore fractal dimension of 2D pore space. The changes are colored in red.

  1. In part 3 It is necessary a deeper explanation of how chemical nature of the solid is taken into account and how different carbonate media can possibly change results. Mention it also in limitation

Response: Special thanks for your good comments. We have made the change according to your valuable comments and suggestions. For example, we have added corresponding content in part 3 Model validation, which is as follows: “In general, carbonates generally contain clay minerals and the interaction between clay minerals and fluids (e.g., water, gas, oil) will significantly affect permeability of porous media (Lei et al., 2019; 2021). For example, Lei et al. (2021) derived analytical permeability to study the effect of clay swelling on permeability of clay-rich argillaceous porous media. In addition, as the interaction between solid minerals (e.g., illite, montmorillonite, and kaolinite) and fluids in porous media (e.g., carbonate media) will change pore structures, the effect of physical or chemical nature of the solid minerals on fluid flow in porous media is of great significance. However, the derived model in this paper does not consider the interaction between solid minerals and fluids. Thus, in our future work, the interaction between solid minerals and fluids will be taken into account to make our model more reasonable.” In addition, we also mention corresponding content in Model advantages and limitations, which is as follows: “Moreover, as mentioned above, the interaction between solid minerals and fluids will significantly affect permeability of porous media. To improve the applicability of our derived model, the interaction between solid minerals and fluids will be taken into account.” And the change is colored in Red.

  1. In lines 374-386 explain if possible, why you are thinking the differences in results. Figure 6D has a similar behavior but 6b present a change in sample 6. why?

Response: Special thanks for your good comments. Figure 6b depicts the pore structure parameters of the eight samples. As pore structures (df, dt, DT, Df and τav) of these samples are different, the curves show different behaviors. However, to further validate our proposed model, the predicted average particle radius from our model is compared with that predicted from former model (Berryman and Blair, 1986; Comiti and Renaud, 1989; Saxena et al., 2017a). Specifically, Figure 6d compares the calculated average particle radius for spherical particles, and the calculated average particle radius for circular particles with the calculated average particle radius from the former model. Results from Figure 6d suggest that the calculated average particle radius from the former model is approximately in the middle between the calculated values from those for spherical particles and circular particles. This indicates that our derived model for average particle radius in porous media gives predicted values which are quite consistent with the results from the former model. In addition, by combining the results from Eqs. (15) and (16), the weight coefficients for different samples have been determined in Figure 6d. Taking samples 3 for example, the weight coefficient is unit, which means average particle radius can be effectively determined by Eq. (C5). We have made the change, which is colored in red.

  1. I strongly recommend include and appendix with symbols, variables and units used in the paper.

Response: Special thanks for your good comments. We have added the Nomenclature in the revised paper according to your valuable comments and suggestions. And the change is colored in Red.

  1. Conclusion needs to consider comments wrote here.

Response: Special thanks for your good comments. We have revised the conclusion as per your valuable suggestions, which is as follows:

Although the model focuses on predicting single phase flow in dry porous media, when the irreducible wetting phase saturation is taken into account, our model can be extended to study the 3D effective permeability of porous media. In addition, it is also available to extend this model to study multi-phase flow in porous media and obtain relative permeability in porous media.

However, it should be noted that our proposed model never considers rough pore surface (e.g., surface fractal dimension) and the interaction in solid minerals-fluid system. Thus, in our future work, more mechanisms will be taken into account to make our model more reasonable.

And the change is colored in Red.

  1. Please explain what is necessary in further work.

Response: Special thanks for your good comments. We have discussed our further work in Model advantages and limitations as per your valuable suggestion, which is as follows: “Furthermore, in general, pore surface of porous materials is rough and surface fractal dimension is crucial to characterize fluid flow in porous media (Lei et al., 2018; Suarez-Dominguez et al. 2020; Xiao et al., 2022). For example, Lei et al. (2018) and Xiao et al. (2022) derived theoretical models to study fluid flow in porous media, and they concluded that the effects of surface fractal dimension on permeability and relative permeability in porous media was significant. Thus, to make our model more reasonable, in our future work, rough pore surface and surface fractal dimension will be taken into account. Moreover, as mentioned above, the interaction between solid minerals and fluids will significantly affect permeability of porous media. To improve the applicability of our derived model, the interaction between solid minerals and fluids will be taken into account.” And the change is colored in Red.
